# An immediate–late gene expression module decodes ERK signal duration

Florian Uhlitz[1,2] (ID), Anja Sieber[1,2], Emanuel Wyler[3], Raphaela Fritsche-Guenther[3], Johannes Meisig[1,2], Markus Landthaler[3,4], Bertram Klinger[1,2] & Nils Blüthgen[1,2,4,*] (ID)

## Abstract

The RAF-MEK-ERK signalling pathway controls fundamental, often opposing cellular processes such as proliferation and apoptosis. Signal duration has been identified to play a decisive role in these cell fate decisions. However, it remains unclear how the different early and late responding gene expression modules can discriminate short and long signals. We obtained both protein phosphorylation and gene expression time course data from HEK293 cells carrying an inducible construct of the proto-oncogene RAF. By mathematical modelling, we identified a new gene expression module of immediate–late genes (ILGs) distinct in gene expression dynamics and function. We find that mRNA longevity enables these ILGs to respond late and thus translate ERK signal duration into response amplitude. Despite their late response, their GC-rich promoter structure suggested and metabolic labelling with 4SU confirmed that transcription of ILGs is induced immediately. A comparative analysis shows that the principle of duration decoding is conserved in PC12 cells and MCF7 cells, two paradigm cell systems for ERK signal duration. Altogether, our findings suggest that ILGs function as a gene expression module to decode ERK signal duration.

**Keywords** ERK signalling; immediate–late genes; mRNA half-life; signal decoding; signal duration
**Subject Categories** Quantitative Biology & Dynamical Systems; Signal Transduction; Transcription
**Mol Syst Biol. (2017) 13: 928**

## Introduction

The RAF-MEK-ERK signalling pathway controls different cellular programmes such as proliferation, differentiation and cell death (Oda *et al*, 2005). It has been shown that these cell fate decisions are encoded by signal duration of its final kinase in the pathway, ERK (Marshall, 1995). Cells commonly interpret transient ERK signalling as a proliferative signal. When exposed to sustained ERK signalling, cells can differentiate or undergo cell death in a cell line-dependent manner. In rat PC12 cells and in human breast cancer MCF7 cells, sustained ERK activity results in cellular differentiation, whereas transient activity elicited by epidermal growth factor (EGF) results in proliferation (Traverse *et al*, 1992; Nagashima *et al*, 2006). Similarly, hamster lung fibroblasts (CCL39) and mouse hippocampal cells (HT22) can discriminate transient and sustained ERK signalling. In both cell types, only prolonged ERK activity accompanied by ERK nuclear retention causes cell death, whereas transient nuclear translocation of ERK is insufficient (Lenormand *et al*, 1998; Stanciu & DeFranco, 2002).

Ultimately, both transient ERK signalling and sustained ERK signalling induce a multitude of early and late responding genes (Amit *et al*, 2007; Tullai *et al*, 2007; Dijkmans *et al*, 2009; Nagashima *et al*, 2009; Saeki *et al*, 2009; Stelniec-Klotz *et al*, 2012). First, primary response genes (PRGs) are induced which in turn mediate expression of secondary response genes (SRGs; Yamamoto & Alberts, 1976). This dependency delays SRG induction and at the same time allows decoding of signal duration, as only prolonged ERK activity ensures sufficient production of required primary factors. In accordance, primary response gene and transcription factor FOS were described to function as a molecular sensor for ERK signal duration (Murphy *et al*, 2002, 2004). When ERK signalling is sustained, FOS protein is stabilised and can promote transcription of specific SRGs. In contrast, when ERK activity is transient, its signal declines before FOS protein can accumulate (Whitmarsh, 2007).

However, the protein sensor model and the concept of PRGs and SRGs cannot explain the ample observation of late primary response genes. When induction of SRGs is blocked with help of the protein biosynthesis inhibitor cycloheximide (CYHX), not only immediate-early primary response genes have been found to be differentially regulated, but also an extensive set of delayed primary response genes (Amit *et al*, 2007; Tullai *et al*, 2007). More precisely, mRNA expression of immediate–early genes (IEGs) peaks within 30–60 min post-EGF stimulation. They are succeeded by delayed–early genes (DEGs), peaking about 120 min post-stimulation (Avraham & Yarden, 2011). Both IEGs and DEGs are primary response genes

1  IRI Life Sciences & Institute for Theoretical Biology, Humboldt Universität Berlin, Berlin, Germany
2  Institute of Pathology, Charité – Universitätsmedizin Berlin, Berlin, Germany
3  Berlin Institute for Medical Systems Biology, Max Delbrück Center for Molecular Medicine, Berlin, Germany
4  Berlin Institute of Health, Berlin, Germany
   *Corresponding author. Tel: +49 30209392390; E-mail: nils.bluethgen@charite.de

(PRGs), as they do not require *de novo* protein biosynthesis. However, composition and RNA dynamics of these temporal gene clusters may differ upon short and prolonged ERK activity, respectively.

Two important questions emerge from this observation. How can primary genes respond late? And how can late primary genes decode signal duration? It has been shown that late induction of gene expression can be caused by restrained splicing of pre-mRNA (Zeisel *et al*, 2011; Hao & Baltimore, 2013; Feldman & Yarden, 2014; Rabani *et al*, 2014) and by long mRNA half-lives (Yang *et al*, 2003; Shalem *et al*, 2008; Hao & Baltimore, 2009; Elkon *et al*, 2010; Nagashima *et al*, 2015; Porter *et al*, 2016; Cheng *et al*, 2017). Interpretation of signal duration on the other hand has been linked to feed-forward regulation of mRNA stability in late response genes (Nagashima *et al*, 2015).

In this study, we present evidence for a new, distinct class of primary response genes that can function in signal duration decoding. Paradoxically, these genes share properties with both IEGs and DEGs. We therefore term them immediate–late genes (ILGs), as they are induced immediately like IEGs, but respond late like DEGs. We demonstrate that long mRNA half-lives dominate mRNA dynamics of ILGs and that this characteristic intrinsically enables them to respond late and to decode signal duration at the same time, without any need for additional regulation. This is in contrast to short-lived IEGs, which do not decode but relay signal duration, postponing the task of duration decoding. The principle of signal duration being translated into response amplitude is conserved in rat PC12 cells and human MCF7 cells, two cell systems which serve as paradigm models for cell fate decisions based on signal duration. In general, mRNA half-life is a strong predictor for response dynamics in these systems. Gene term enrichment analysis furthermore proposes a potential role of ILGs in positive regulation of apoptosis. As IEGs are found to be involved in negative regulation of apoptosis, we speculate that the two opposing modules together could serve as a fail-safe mechanism upon prolonged versus transient ERK signalling.

# Results

## A highly controllable synthetic cell culture system allows modelling of ERK downstream targets

Signals received at the cell surface propagate through a network of signalling proteins (Oda *et al*, 2005; Kholodenko *et al*, 2010). Eventually, signalling events activate complex gene expression programmes. However, it is difficult to address how signalling input dynamics are translated into gene expression output dynamics, as different signalling pathways can be activated by the same receptor and different receptors can activate the same pathway (Kholodenko *et al*, 2010). Activated pathways can intertwine, counteract each other or converge on the same downstream promoters (Parikh *et al*, 2010). Negative feedback loops allow for adaptation to constant signal exposure and complicate the attempt to link gene expression programmes to signalling inputs even further. In epidermal growth factor receptor (EGFR) signalling, the activated RAF-MEK-ERK and PI3K-AKT signalling pathways cross talk extensively (Mendoza *et al*, 2011;

Aksamitiene *et al*, 2012; Fritsche-Guenther *et al*, 2016). They share downstream transcription factors (Tullai *et al*, 2004) and a multitude of feedback loops act on both pathways (Kolch *et al*, 2015). To distinctively characterise the gene expression program specifically elicited by ERK signalling, we hence used a synthetic cell culture system that allowed for tight control of ERK activity. In contrast to growth factor-induced systems, this ERK signalling model system is uncoupled from the upstream G protein RAS, avoiding pathway divergence and feedback mechanisms.

More precisely, we used human embryonic kidney (HEK293) cells constitutively expressing an inducible form of RAF ($\Delta$RAF1:ER; Samuels *et al*, 1993; McMahon, 2001; Cagnol *et al*, 2006; Fig 1A) to generate a wide range of different gene expression time course data sets (Fig EV1). It was previously reported that constant exposure to estrogen receptor (ER) antagonist 4-hydroxytamoxifen (4OHT) causes sustained activation of ERK signalling and results in Caspase-8-mediated induction of apoptosis in these cells, whereas parallel treatment with small molecule MEK inhibitor U0126 does not (Cagnol *et al*, 2006). In addition, the system mimics oncogenic RAF signalling and therefore provides insights into early onset of RAF-driven malignancies and the decisive competition between anti-apoptotic and pro-apoptotic signals elicited by the RAF-MEK-ERK signalling network. In this work, we used this synthetic system to generate pulses of ERK signalling with defined duration, by stimulating RAF activity with 4OHT and subsequently blocking it using U0126. Using transcriptomics time series, it serves as an excellent model system to study mRNA dynamics of downstream targets upon different ERK signalling durations.

Gene expression time course data unveiled sustained induction of 189 target genes (at FDR = 1%) upon constant exposure to 4OHT (ON scenario, Fig 1B). In contrast, subsequent inactivation with U0126 two hours post-induction resulted only in transient induction of these target genes (ON/OFF scenario). To distinguish primary and secondary response genes, protein biosynthesis inhibitor cycloheximide (CYHX) was applied in parallel to 4OHT treatment. 102 genes were still induced upon parallel CYHX treatment (ON/CYHX scenario) and considered primary response genes (PRGs). The remaining 87 genes were no longer significantly induced in the presence of CYHX and therefore not considered PRGs. However, a multitude of different mRNA dynamics was observed among PRGs with immediate, delayed and late responses. As elaborated above, these transcriptional waves have been termed immediate–early genes (IEGs) and delayed–early genes (DEGs). So far, classification of IEGs and DEGs has only been based on peak expression time points (Amit *et al*, 2007; Tullai *et al*, 2007). In this study, we based distinction of IEGs and DEGs on mathematical modelling. This approach allowed us to quantify transcriptional delays in our experimental data and to distinguish temporal gene clusters in a sustained signalling scenario where peak expression cannot be defined.

## Mathematical modelling of mRNA dynamics

We based our mathematical model on a minimal model of gene expression (Gorini & Maas, 1957) with basal transcription rate $k_0$ and degradation rate $\gamma$. We extended the minimal model with two additional parameters to account for dynamics in ERK activity and

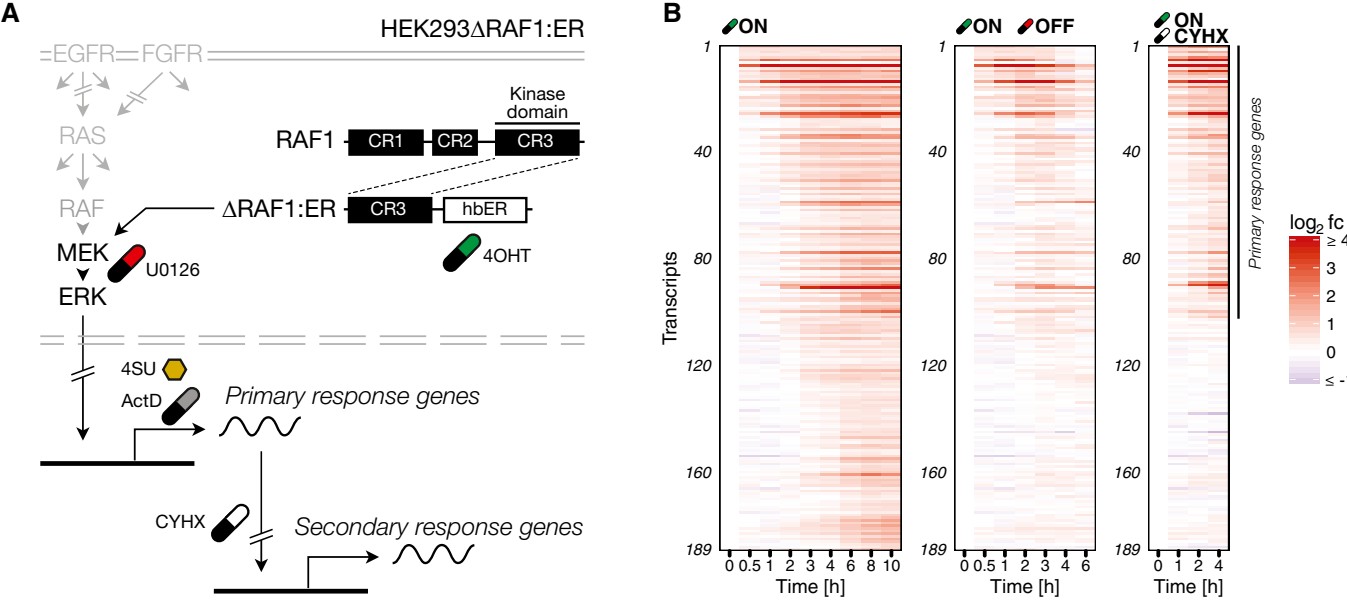

**Figure 1.  Expression kinetics from a synthetic model system for ERK signal duration.**

A   HEK293 with a stably transfected ΔRAF1:ER fusion protein were treated with ER antagonist 4-hydroxytamoxifen (4OHT, ON scenario in B). To generate pulses, ERK signalling was turned off using the MEK inhibitor U0126 (ON/OFF scenario in B). To distinguish between primary and secondary response genes, translation was blocked with cycloheximide (CYHX) in parallel to 4OHT stimulation (ON/CYHX scenario in B). In addition, we used actinomycin D (ActD) to determine mRNA half-lives via transcriptional shutdown and 4-thiouridine (4SU) to determine mRNA half-lives via metabolic labelling.

B   Log₂ gene expression fold changes of significantly induced genes (FDR = 1%) across different treatment scenarios. Gene induction of immediate, delayed and late responding genes is sustained upon constant activation (ON scenario) and transient upon two-hour pulse activation (ON/OFF scenario). Genes still significantly induced upon parallel CYHX treatment were considered primary response genes. Genes were ranked by their model-derived response time.

to model transcriptional delays. Mathematically, we allowed for an additional ERK activity-dependent transcription rate $k$ and a delay parameter $\Delta t$:

$$\frac{\mathrm{d}[mRNA](t)}{\mathrm{d}t} = k_0 + k \cdot pERK(t - \Delta t) - \gamma[mRNA](t)$$

Biologically, the additional delay parameter $\Delta t$ here accounts for all steps that need to take place before transcription can start, like chromatin remodelling, transcription factor recruitment and polymerase recruitment.

### *In silico* analysis predicts that RNA half-lives and transcriptional delays shape mRNA dynamics

Sustained, pulsed and transient signalling kinetics elicit different mRNA dynamics (Fig 2A). But mRNA dynamics are not solely determined by signalling inputs. It is different combinations of RNA half-lives and transcriptional delays that enlarge the number of possible gene expression profiles. Mathematical modelling of RNA production and degradation suggests (Yang *et al*, 2003) and experimental data confirm (Shalem *et al*, 2008; Hao & Baltimore, 2009; Elkon *et al*, 2010; Nagashima *et al*, 2015; Porter *et al*, 2016; Cheng *et al*, 2017) that short-lived transcripts can be induced more rapidly than long-lived transcripts. In consequence, pulse or transient signalling inputs are sufficient for induction of short-lived mRNAs, whereas long-lived mRNAs require sustained signalling inputs to exceed their half maximum response amplitude (Fig 2A). In addition, transcriptional delays

can shift gene induction, leading to expression peaks of target genes hours after signal inputs lapsed (Fig 2A).

*In silico* exploration of RNA half-life and transcriptional delay parameter space showed that immediate–early genes (characterised by short half-life and short transcriptional delay) consistently reach at least 50% of their response amplitude in all simulated input scenarios (sustained, pulse, transient, Fig 2B). Likewise, simulations predicted that short-lived delayed–early genes are also capable to exceed this level of response amplitude, but in a delayed fashion. Lastly, simulations confirmed that only long-lived mRNAs require sustained signalling to reach at least 50% of their response amplitude.

It is important to note that response amplitudes are presented as relative values normalised to steady-state expression. Such normalised values ease the comparison of the timing between different genes during their transition from one steady state to another. At the same time however, this representation cannot reflect absolute changes in mRNA concentration. Hence, we present relative changes in expression (noted as amplitude [%]) when describing the relation between mRNA half-life and signal duration decoding and absolute changes in expression (noted as log₂ fold change), when we focus on quantitative aspects of mRNA expression.

Altogether, simulations suggested that both short-lived and long-lived transcripts can discriminate different signal durations. However, only long-lived genes truly decode signal duration by translating it to response amplitude, whereas short-lived genes relay signal duration to response duration, postponing the task of decoding.

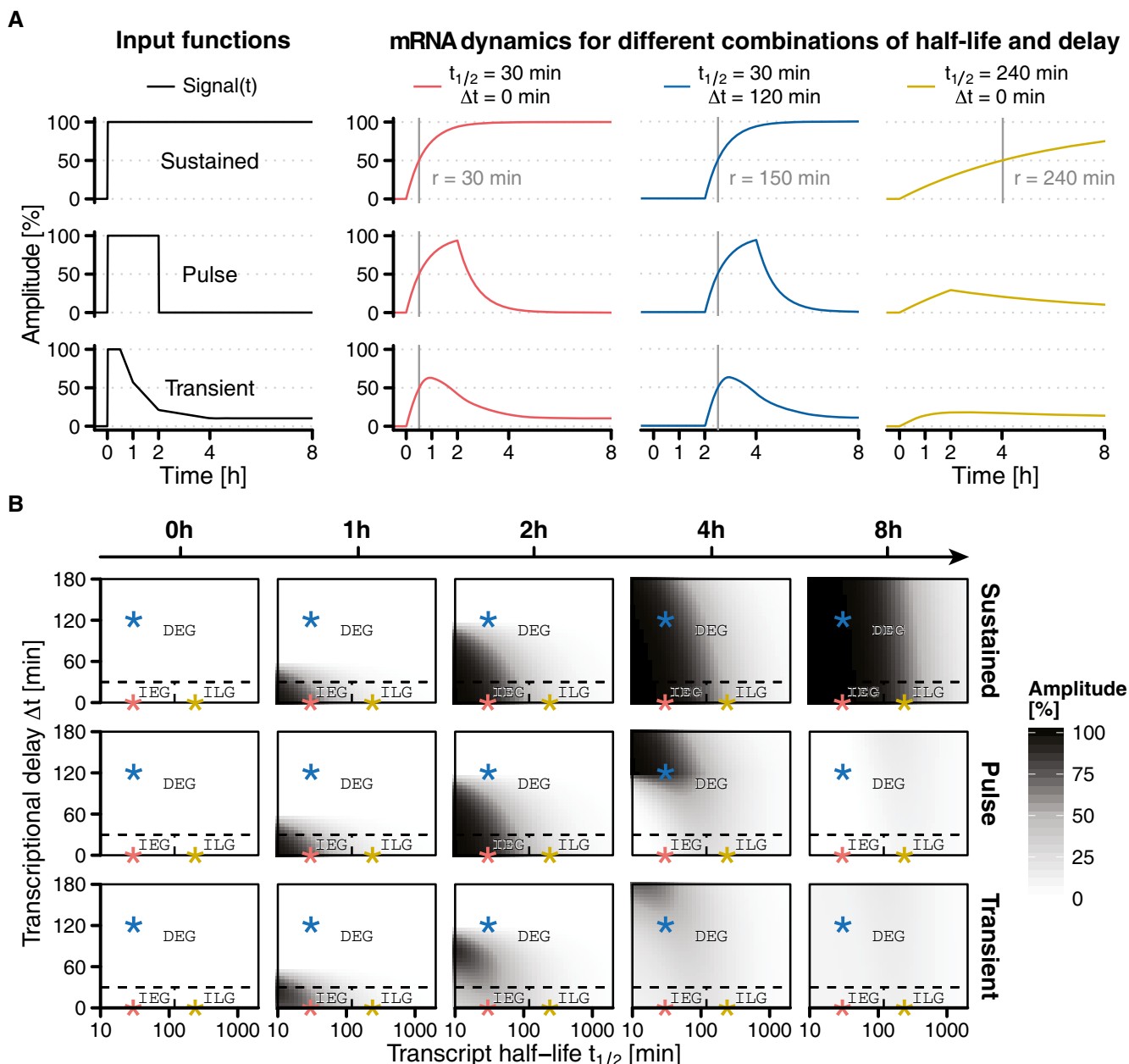

**Figure 2. Simulation of primary response gene dynamics upon different signalling durations.**

A  Different activation patterns of signalling molecules (input functions, left) can elicit multiple different response profiles (right) with different response times ($r$) depending on mRNA half-lives ($t_{1/2}$) and transcriptional delays ($\Delta t$). Rapid induction requires short half-lives (red lines). Late induction can be caused by transcriptional delays (blue lines), long half-lives (yellow lines) or combinations thereof. Decoding of signal duration depends on mRNA half-life. Short-lived mRNAs relay signal duration to response duration, whereas long-lived mRNAs decode signal duration to response amplitude (yellow lines).

B  Response amplitude for all simulated combinations of mRNA half-life and transcriptional delay. Response amplitude is shown over time (columns) and in respect to input function (rows). For sustained signalling, all primary response genes exceed their half maximum response amplitude. Pulse and transient signalling inputs are only sufficient for immediate–early genes and short-lived delayed–early genes. Long-lived mRNAs with half-lives greater 120 min require sustained signalling inputs to exceed their half maximum response amplitude. Example parameter sets displayed in (A) are marked with asterisks in (B). Dashed lines indicate cluster borders. IEG: immediate–early genes, $t_{1/2} \leq 120$ min and $\Delta t \leq 30$ min. ILG: immediate–late genes, $t_{1/2} > 120$ min and $\Delta t \leq 30$ min. DEG: delayed–early genes, $\Delta t > 30$ min.

### Identification of IEGs, DEGs and a new temporal cluster of immediate–late genes (ILGs)

To train our mathematical model of gene expression and to infer gene-wise model parameters for all induced primary response genes, we used the ON condition as a training set. For this, we integrated stimulus-dependent phosphorylation of ERK measured in a multiplex immunoassay (Bio-Plex) and gene expression data obtained from Affymetrix Human Gene 1.0 ST microarray time course experiments (Fig 3A).

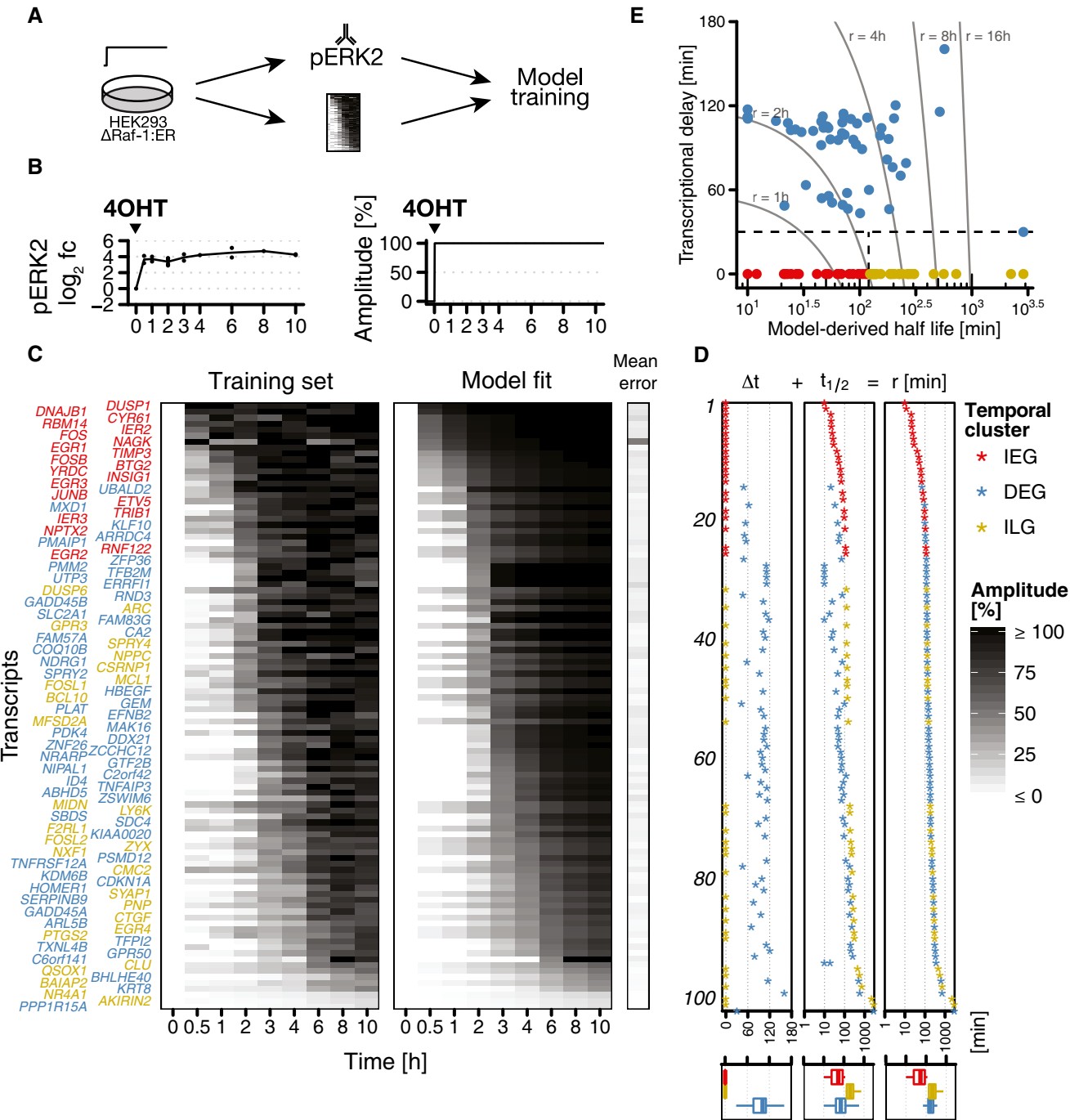

**Figure 3.  Model fitting and classification of primary response genes.**

A  HEK293ΔRAF1:ER cells were treated with 4OHT for constitutive induction of ERK signalling. Phosphorylation levels of ERK2 were measured with bead-based ELISA (Bio-Plex). RNA time course expression data were measured using microarrays. These data were the basis to train gene-wise model parameters.

B  pERK2 log$_2$ fold change upon sustained activation (left) and deduced input function (right) used for model fitting. Average pERK2 log$_2$ fold change upon 4OHT treatment equals 100% signalling amplitude.

C  Measured gene expression kinetics and the resulting maximum-likelihood fit of the gene expression model of all significantly induced primary response genes (FDR = 1%). Gene expression is shown as percentage of response amplitude. Mean error is calculated as the mean of absolute residuals serving as a goodness of fit measure.

D  Classification of primary response genes. For each gene, response time $r$ is calculated as the sum of deduced transcriptional delay $\Delta t$ and mRNA half-life $t_{1/2}$ and used for ranking. In general, immediate–early genes (IEGs) have short response times, whereas both immediate–late genes (ILGs) and delayed–early genes (DEGs) have longer response times. For ILGs, long response times are due to long mRNA half-lives. For DEGs, long response times are mainly due to long transcriptional delays (cf. boxplots).

E  Response times for IEGs, DEGs and ILGs. Genes on the same trajectory have the same response time, but response times are composed differently. For IEGs and ILGs, response times are solely determined by mRNA half-life, whereas DEGs have response times that are mixtures of half-life and transcriptional delay.

Source data are available online for this figure.

Across all treatment durations for the ON condition, pERK was reliably induced with mean $\log_2$ fold change of $3.89 \pm 0.42$ (Fig 3B). We therefore deduced a simplified input function of $pERK(t) = 1$ for $t > 0$ and incorporated it into our extended model of gene expression. We then fitted basal transcription rate $k_0$, pERK-dependent transcription rate $k$, degradation rate $\gamma$ and transcriptional delay $\Delta t$ for each of the 102 significantly induced primary response genes considering an error model to account for expression level-dependent variance (Fig 3C). In a simplified model, we left out transcriptional delay parameter $\Delta t$ and fitted all remaining parameters again. Using a likelihood ratio test (Kreutz & Timmer, 2009), we compared the sum of weighted squared residuals (wRSS) of the complete and simplified model (Fig EV2). The complete model was only accepted for genes with significantly enhanced fits (*P*-value <0.05). For the remaining genes and for genes with $\Delta t < 30$ min (to reflect sampling intervals), half-life estimates ($t_{1/2} = \ln(2)/\gamma$) were based on the simplified model. All fitted parameter values are listed in Table EV1.

Genes were ranked according to their model-derived response time $r$, which was calculated as the sum of model-derived half-life and transcriptional delay ($r = \Delta t + t_{1/2}$, Fig 3D). For each gene, the response time $r$ corresponds to the time when it reaches its half maximum response amplitude. Based on this, we identified 21 rapidly induced IEGs with median response time of 53 min. About half of all induced PRGs (54/102) were classified as DEGs with $\Delta t > 30$ min and median response time of 160 min. Lastly, we identified 27 immediate–late genes with median response time of 204 min and half-lives greater 120 min.

Scatter plotting of half-lives and transcriptional delays allowed visualisation of response time composition, the temporal order of gene clusters (Fig 3E). IEGs defined the first wave of response with model-derived half-lives ranging from 10 (*DUSP1*) to 117 min (*EGR2*). Both DEGs and ILGs subsequently responded after several hours (2–48 h) with few exceptions (6 DEGs showed $r < 2$ h: *UBALD2, MXD1, KLF10, ARRDC4, PMAIP1, ZFP36*). DEGs showed median half-life of 70 min and median transcriptional delay of 102 min. Responding ILGs showed half-lives ranging from 124 min (*DUSP6*) up to 561 min (*QSOX1*). For one DEG (*PPP1R15A*) and three ILGs (*BAIAP2, NR4A1, AKIRIN2*), model-derived response times were >10 h, the time span covered in the experiment. All summarising values assigned to particular gene clusters like here need to be considered bearing in mind the continuous nature of gene expression parameters apparent in our analysis (Fig 3D and E).

## Model-derived parameters allow semi-quantitative prediction of mRNA $\log_2$ fold changes elicited by different signalling scenarios

In a next step, we could demonstrate that the parameter knowledge we gained about PRGs induced upon sustained ERK signalling can be used to semi-quantitatively predict their behaviour upon different signalling scenarios (Fig 4). We compared sustained ERK signalling elicited by 4OHT with a 2-h pulse (4OHT followed by U0126) and with EGF and fibroblast growth factor (FGF) treatment. Bead-based ELISAs confirmed that these stimuli indeed result in sustained, 2-h pulse and native growth factor-induced pERK dynamics, respectively (Fig 4B). EGF caused transient ERK activation (max. $\log_2$ fold change: $5.43 \pm 1.05$) and FGF caused attenuated but sustained ERK activation (mean $\log_2$ fold change: $2.83 \pm 0.50$) (Fig 4B). Growth factor-mediated input functions required for prediction were generated from linear interpolations of pERK2 $\log_2$ fold changes relative

to mean induction in test condition. Deduced input functions were then incorporated into our mathematical model of gene expression.

Strikingly, trained parameters allowed for semi-quantitative prediction of mRNA $\log_2$ fold changes upon two-hour pulse signalling as well as upon EGF and FGF treatment (Fig 4C). Mean relative prediction error was $18.1\% \pm 11.8\%$ for 2-h pulse, $19.1\% \pm 15\%$ for EGF and $20.5\% \pm 12.3\%$ for FGF (cf. Fig EV3 for cluster-wise boxplots of mean prediction errors). The predictive power of the model trained on the on-kinetics underlined that a minimal model of gene expression is sufficient to describe mRNA dynamics in our cell system for ERK signal duration with high accuracy.

## Transcriptional shutdown data confirms short mRNA lifespan of IEGs and longevity of ILGs

To verify our half-life estimates based on modelling of gene induction for the different temporal gene clusters, we determined mRNA half-lives upon actinomycin D (ActD)-mediated transcriptional shutdown and upon metabolic labelling of RNA with 4-thiouridine (4SU) (Fig 5A, cf. Fig EV1 for sampling). ActD-derived half-lives were determined in both 4OHT-pretreated and untreated HEK293ΔRAF1: ER cells (Fig 5A, ON and OFF panels). 4SU-derived half-lives were only determined in untreated HEK293ΔRAF1:ER cells (Fig 5A, OFF panel), since this approach assumes steady-state gene expression. In general, ActD-derived mRNA half-lives in 4OHT-pretreated and untreated samples correlated well (Fig EV4A, Spearman's rho = 0.74). Also, estimates from the two different methods correlated well (Fig EV4B, Spearman's rho = 0.57). Moreover, median mRNA half-life estimates based on all three data sets showed good correlation with published data on human mRNA half-life [Fig EV4C, Spearman's rho = 0.60 with Friedel *et al* (2009) and rho = 0.66 with Yang *et al* (2003)]. All mRNA half-life estimates are provided as supplementary data in Table EV2.

ActD-derived mRNA half-life estimates from 4OHT-pretreated HEK293ΔRAF1:ER cells confirmed short mRNA half-lives for IEGs (median = 60 min) and longer mRNA half-lives for ILGs (median = 230 min, Fig 5A, ON panel). Data from untreated cells yielded highly reproducible half-life estimates for uninduced genes, but resulted in different half-life estimates for induced genes (Fig 5A, OFF panel). Most prominently, mRNA half-life estimates for all IEGs were smaller in 4OHT-pretreated than in untreated data. This could hint either at destabilisation of IEGs in response to ERK signalling or at difficulties in determination of half-lives from transcriptional shutdown experiments for induced genes under baseline conditions.

Notably, all three independent measurements identified half-lives for DEGs in range of ILGs (> 120 min), whereas estimates based on modelling of gene induction had suggested shorter half-lives for DEGs in range of IEGs (< 120 min). We therefore concluded that half-lives of IEGs and ILGs can certainly be estimated from gene induction kinetics, whereas half-lives of DEGs are more reliably determined in transcriptional shutdown experiments.

## ILGs are transcribed immediately and have GC-rich promoters like IEGs

Our mathematical model predicted that ILGs respond late like DEGs, but are induced immediately like IEGs. It has been reported

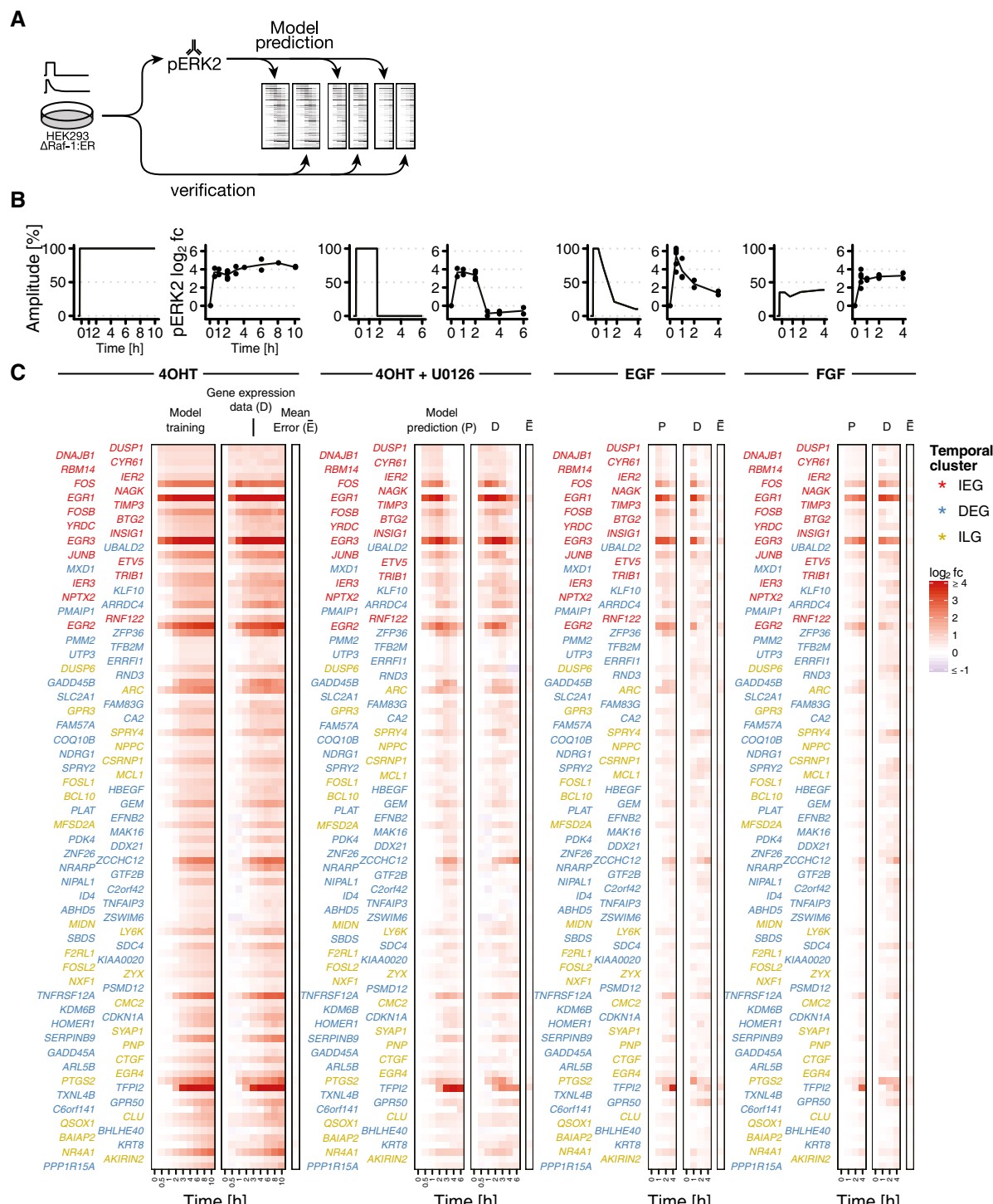

**Figure 4. Semi-quantitative prediction of mRNA log₂ fold changes upon different signalling scenarios.**

A    Gene expression upon different stimulations was predicted based on fitted model parameters and measured pERK2 levels. Predictions were verified with gene expression time course data.

B    Signalling input conditions (left side shows deduced input function, and right side shows pERK2 measurements): Sustained ERK signalling (4OHT), 2-h pulse ERK signalling (4OHT + U0126), growth factor signalling (EGF: epidermal growth factor, FGF: fibroblast growth factor). Deduced input functions: 100% signalling amplitude corresponds to mean induction in training condition (4OHT). Growth factor-induced input functions are linear interpolations of pERK2 log₂ fold changes relative to mean induction in test condition.

C    Predictions are verified with actual gene expression data. Heat maps show log₂ fold changes of induced mRNAs. P: model prediction. D: gene expression data. E: mean error = mean of absolute residuals.

Source data are available online for this figure.

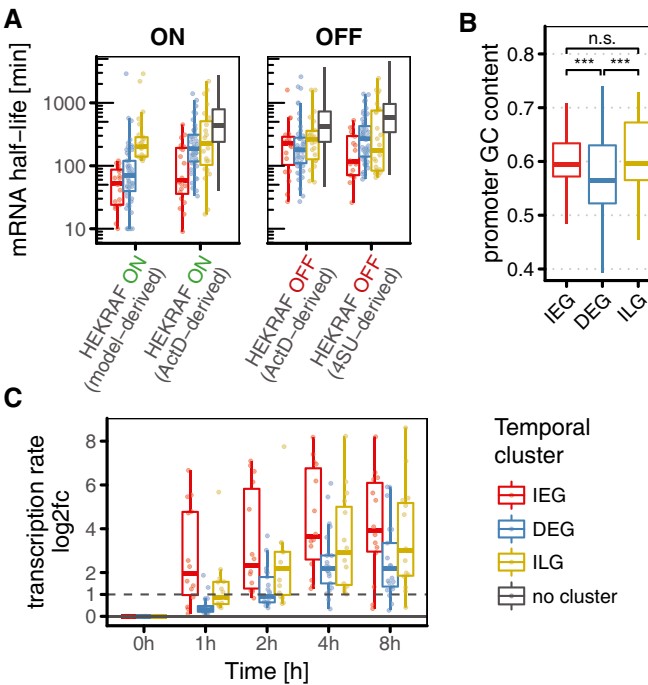

**Figure 5. Immediate–late genes (ILGs) have long mRNA half-lives, are transcribed immediately and have GC-rich promoters.**

A  Boxplot comparison of mRNA half-life estimates based on modelling of gene induction (model-derived), transcriptional shutdown (ActD-derived) and metabolic labelling (4SU-derived). Estimates from 4OHT-pretreated HEK293ΔRAF1:ER cells (ON panel) are more appropriate to characterise induced genes than estimates from unstimulated cells (OFF panel). Genes not assigned to any cluster are shown in grey.

B  Promoter GC content in IEGs, DEGs and ILGs. Calculated for TSS ± 1,000 bp in hg19. Wilcoxon rank sum was used to check for significant differences (n.s.: not significant, ***: P-value < 0.001).

C  Log$_2$ fold changes of transcription rate in 4OHT-treated HEK293ΔRAF1:ER cells derived from metabolic labelling (4SU) RNA-sequencing data document immediate transcription of IEGs and ILGs but delayed transcription of DEGs. Dashed horizontal line indicates doubling of transcription rate.

Data information: Boxplots show median and inter-quartile range. IQR is extended with whiskers to the largest and smallest value respectively, but no further than 1.5× IQR from hinges.

previously that IEGs and DEGs differ in their promoter architecture (Tullai *et al*, 2007; Ramirez-Carrozzi *et al*, 2009; Avraham & Yarden, 2011). IEGs commonly possess GC-rich promoters, allowing for instant activation independent of nucleosome remodellers, whereas genes that respond with a delay commonly possess GC-poor promoters, facilitating their dependence on remodellers and thereby delaying their induction (Ramirez-Carrozzi *et al*, 2009). Given our model-based classification of temporal gene clusters, we could confirm that IEGs show GC-rich promoters (TSS ± 1,000 bp), and DEGs show GC-poor promoters (Fig 5B). Based on the model, we hypothesised immediate transcription of ILGs and that therefore their promoters should be similar to those of IEGs. And indeed, we found that ILG promoters (TSS ± 1,000 bp) are likewise GC-rich. Their GC content is in the same range as of IEG promoters, and significantly higher (Wilcoxon *P*-value = $1.4 \times 10^{-4}$) than GC content of DEG promoters (Fig 5B). Hence, immediate induction

of ILGs is potentially facilitated by their permissive promoter architecture.

To directly confirm that ILGs are immediately induced, we used 1-h pulses of metabolic labelling with 4SU followed by RNA sequencing and compared changes in nascent transcripts as proxy of the transcription rate across clusters. Indeed, median transcription rate of ILGs was approximately doubled just one hour after induction of ERK signalling with 4OHT (Fig 5C), suggesting immediate transcription. DEGs in contrast required 2 h of 4OHT treatment until median transcription rate was doubled, suggesting delayed transcription. Overall, IEGs showed the steepest changes in transcription rate. As mRNA levels are determined by the ratio of mRNA production and decay rate, this suggests that IEGs compensate their short half-lives with high transcription rates and may therefore reach similar steady-state levels as ILGs. Indeed, absolute transcription rate (TPM/h = transcripts per million per hour) showed anti-correlation with model-derived half-life estimates for IEGs and ILGs (Fig EV5A). Comparison of nascent mRNA and total mRNA levels further confirmed that IEGs can reach high total mRNA levels after both short and prolonged ERK activity (total RNA in Fig EV5B), by compensating their short half-lives with very high transcription rates (nascent RNA in Fig EV5B). Interestingly, total mRNA levels further showed that ILGs are low expressed before stimulation, but reach similar levels as IEGs and DEGs after prolonged activation (total RNA in Fig EV5B).

## ILGs translate ERK signal duration into response amplitude

Having quantitatively characterised and validated the parametric properties of IEGs, DEGs and ILGs, we moved on to validate our main hypothesis, the capacity of ILGs to translate ERK signal duration into response amplitude. First, we compared response amplitudes upon constant ERK signalling elicited by 4OHT with response amplitudes upon two-hour pulse (4OHT followed by U0126) and transient signalling (EGF treatment) in our transcriptome time course data (cf. Fig EV1A for sampling).

Median induction curves of the different proposed PRG clusters nicely reflected their different kinetic properties (Fig 6A). Again, the rapid induction of IEGs and the subsequent response of DEGs and ILGs were apparent. However, overall accumulation of DEGs was delayed first but rapid later, whereas overall accumulation of ILGs was immediate but steadily slow. Remarkably, when considering shortened ERK signal durations, some genes were still rising in expression hours after signalling inputs lapsed (Fig 6A). This behaviour was predominantly observed among DEGs and more pronounced upon EGF-mediated ERK signalling than upon two-hour pulse signalling. In contrast, expression of IEGs and ILGs nearly instantly declined once signalling inputs lapsed. This observation was very much in accordance with our *in silico* data (Fig 2).

When systematically comparing the capacity of IEGs and ILGs to decode signal duration based on this data, we saw that IEGs generally relayed signal duration to response duration. Response amplitude however was only partially affected for a fraction of IEGs. For all IEGs, a two-hour pulse was sufficient to exceed 50% of response amplitude (Fig 6B). Remarkably, even transient activation with EGF was sufficient to induce half of IEGs (11/21) to this extend.

In contrast, response amplitude of ILGs was strongly linked to ERK signal duration. Upon sustained signalling, the majority of ILGs

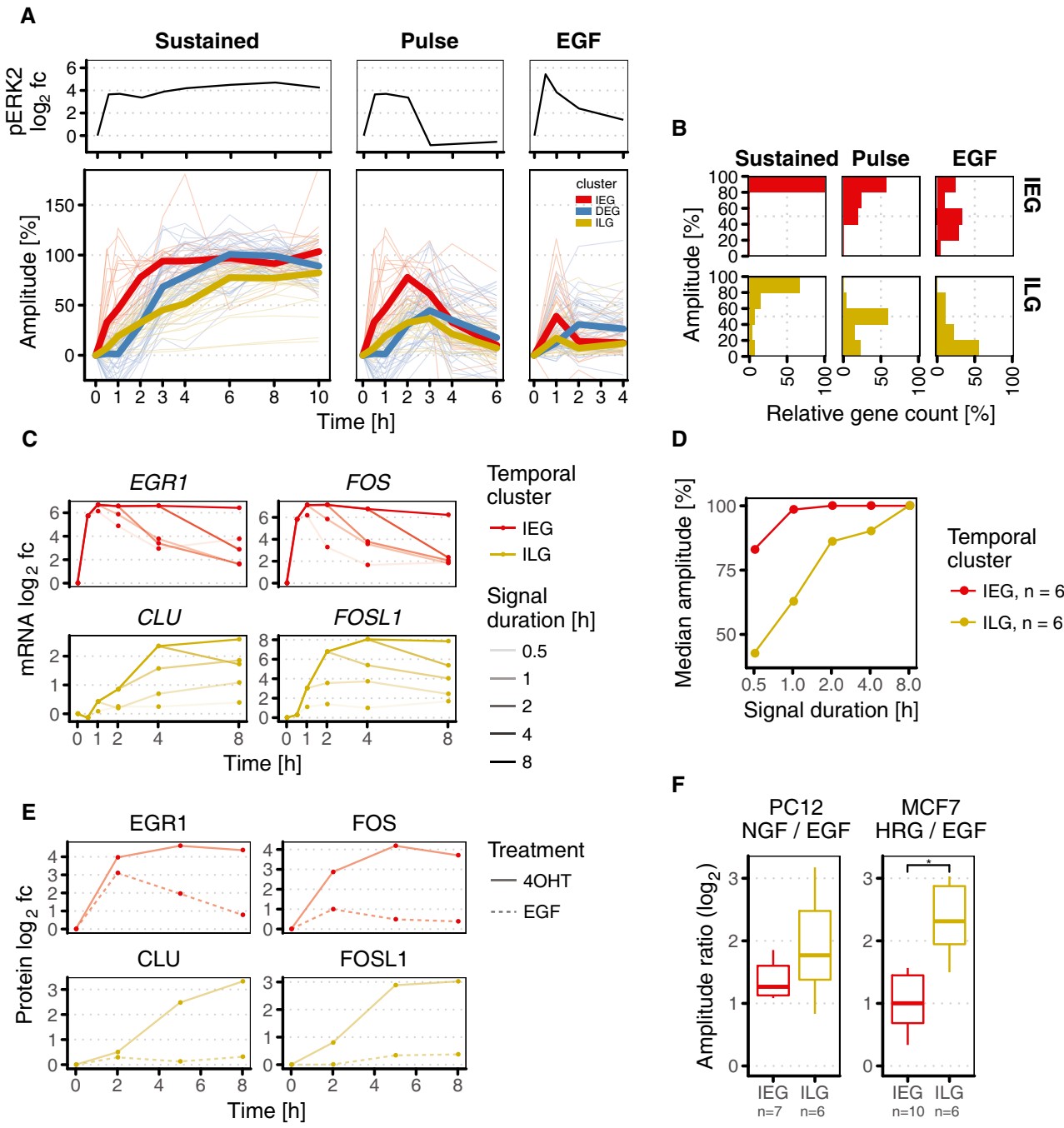

**Figure 6. Immediate–late genes (ILGs) translate signal duration into response amplitude.**

A    Upper panel: pERK2 log$_2$ fold changes upon different input scenarios (sustained: 4OHT; 2-h pulse: 4OHT + U0126). Lower panel: response amplitude across temporal clusters and signal durations. Bold lines show median cluster amplitude at each time point.

B    Capacity to decode signal duration. ILGs translate ERK signal duration into response amplitude. IEGs are only partially able to do so, as many of these genes are still strongly induced upon shortened signal durations.

C    qPCR validation to test different ERK signal durations. HEK293ΔRAF1:ER cells were treated with 4OHT and U0126 for different periods of time to generate signal duration scenarios of 0.5–8 h (cf. Fig EV1C). mRNAs of IEGs EGR1 and FOS relay signal duration to response duration, whereas ILGs CLU and FOSL1 decode signal duration to response amplitude (qPCR data for all 17 validated mRNAs is shown in Fig EV6A).

D    Relation between signal duration and response amplitude for IEGs and ILGs derived from qPCR validation data. Median amplitude is based on six qPCR-validated ILGs and six qPCR-validated IEGs.

E    Quantification of Western blots to present protein log$_2$ fold changes of sample genes upon sustained ERK signalling (4OHT-induced) and transient ERK signalling (EGF-induced) in HEK293ΔRAF1:ER cells.

F    Conservation of signal duration decoding to response amplitude in two prominent model systems for ERK signal duration: NGF and HRG cause more sustained ERK signalling compared to EGF treatment in PC12 and MCF7 cells, respectively. Decoding of signal duration to response amplitude is clearly governed by ILGs.

reached response amplitudes between 80 and 100%, similar to IEGs. However, upon two-hour pulse signalling, the majority of ILGs reached response amplitudes only between 40 and 60%, and upon transient EGF-mediated signalling, the majority of ILGs did not exceed 20% response amplitude. Thus, ILGs clearly distinguished sustained and short signalling by translating signal duration into response amplitude. Supporting our hypothesis that long mRNA half-lives enable this decoding mechanism, we found that a fraction of long-lived DEGs with model-derived half-lives greater 120 min was also capable of decoding ERK signal duration in a similar manner (*ARL5B, BHLHE40, C2orf42, CDKN1A, GADD45A, GPR50, HOMER1, KDM6B, KRT8, PPP1R15A, SERPINB9, TFPI2, TNFRSF12A, TXNL4B*).

Next, we investigated in detail the relation between signal duration and response amplitude in a qRT–PCR expression panel consisting of seventeen highly regulated IEGs, ILGs and long-lived DEGs (Fig EV6A). Using our synthetic system, we activated ERK signalling for five different durations ranging from half an hour to eight hours, and measured the time kinetics of these seventeen genes (cf. Fig EV1C for sampling). The resulting time series shows that IEGs like *EGR1* and *FOS* relay signal duration to response duration, that is longer pulses lead to longer expression with the same amplitude (Fig 6C). ILGs like *CLU* and *FOSL1* on the contrary decode signal duration by translating it into response amplitude, that is maximal expression increased with longer signal duration. The principle was not only apparent when looking at individual genes, but also in the average response amplitude for all qRT–PCR-validated genes (Fig 6D). For IEGs, the median amplitude of IEGs increases only slightly from 0.5 to 1 h and remains at 100% for longer durations. In contrast, the median amplitude of ILGs increases steadily with higher signal duration. Taken together, these data confirm that IEGs relay signal duration, whereas ILGs decode signal duration into response amplitude.

We were also interested if such decoding is also apparent at the protein level and tested the proteins EGR1, FOS, CLU and FOSL1 (Fig 6E). For EGR1, we find that also the protein relays signal duration but does not decode it. FOS protein levels, in contrast, did show strong duration decoding. This is in accordance with the literature (Murphy *et al*, 2002, 2004), as the FOS protein is stabilised in an ERK-dependent fashion. Therefore, IEGs can decode signal duration on the protein level if the protein itself is stable or stabilised (Fig 6E). For both tested ILGs CLU and FOSL1, we find that also their proteins show strong differences in response amplitude for the different input stimuli, and can therefore decode signal duration.

### ILG decoding of ERK signal duration is conserved in PC12 and MCF7

So far, our analyses were focussed on a synthetic cell culture system that allows for precise control of ERK signal duration. To provide evidence that the principle of how ILGs decode signal duration is conserved in more physiological conditions, we examined two other paradigm model systems for ERK signal duration. As elaborated above, PC12 cells undergo proliferation or differentiation when exposed to transient or sustained ERK signalling elicited by EGF or neuronal growth factor (NGF), respectively. Likewise, MCF7 cells undergo proliferation or commit to apoptosis when exposed to

transient or sustained ERK signalling elicited by EGF or heregulin (HRG), respectively. We calculated amplitude ratios for significantly induced IEGs and ILGs in MCF7 and for homologues in PC12 using publicly available data sets (Shiraishi *et al*, 2010; Offermann *et al*, 2016). Both NGF/EGF amplitude ratio in PC12 and HRG/EGF amplitude ratio in MCF7 were higher for ILGs when compared to IEGs.

To further provide evidence that mRNA half-lives dominate gene expression timing and hence translation of signal duration into response amplitude, we compared genes induced in both 4OHT-treated HEK293ΔRAF1:ER and PC12 or MCF7 cells, respectively. Whereas maximum $\log_2$ fold changes showed poor correlation, both our model-derived response times and median mRNA half-life estimates correlated nicely with peak expression time points in both PC12 and MCF7 (Spearman's rho between 0.60 and 0.70, Fig EV6C and D). This suggested that gene expression timing is highly conserved in all three tested models for ERK signal duration.

### ILGs might serve as a fail-safe mechanism to control aberrant ERK signalling in HEK293

Previous studies have suggested that IEGs and DEGs are distinct in function and that they can regulate each other. Whereas IEGs have been described as feed-forward elements predominantly encoding transcription factors (boosting the expression of DEGs and inducing the expression of SRGs), DEGs have been described as a module of negative feedback regulators (Amit *et al*, 2007; Tullai *et al*, 2007; Kholodenko *et al*, 2010; Avraham & Yarden, 2011). These negative feedback regulators include phosphatases (DUSPs), which inactivate MAP kinases (Fritsche-Guenther *et al*, 2011); RNA binding proteins, which mediate degradation of IEGs (e.g. ZFP36, which binds FOS mRNA); and other negative feedback elements, such as tumour suppressors (Amit *et al*, 2007).

To check whether our classification of different PRG subclasses is consistent with known functional annotations, we performed Gene Ontology (GO) term enrichment analysis (Fig EV7A). In accordance with the literature, IEGs were enriched for positive regulators of transcription from PolII promoter (9 out of 21: *CYR61, EGR1, EGR2, ETV5, FOS, FOSB, INSIG1, JUNB, RBM14*). Among induced DEGs, we identified aforementioned RNA binder *ZFP36*, negative receptor feedback elements *ERRFI1* and *SPRY2*, and other negative feedback regulators of protein kinase activity (*GADD45A, GADD45B, CDKN1A, TNFAIP3*). Moreover, the top upregulated DEG was tumour suppressor tissue factor pathway inhibitor 2 (*TFPI2*).

Having confirmed the different functional roles of IEGs and DEGs, we moved on to functionally characterise our newly defined gene cluster of ILGs. Strikingly, GO term enrichment analysis suggested a distinct role of ILGs in positive regulation of apoptosis, putatively opposing involvement of IEGs in negative regulation of apoptosis. This finding suggests that the capability of ILGs to decode ERK signal duration might serve as a potential fail-safe mechanism to control aberrant ERK signalling, as these positive regulators of apoptosis only come into play, when ERK is activated in a prolonged fashion.

In general, it has been shown that RAF-MEK-ERK signalling is involved in positive and negative regulation of both intrinsic (mitochondrial) and extrinsic (receptor) pathway of apoptosis (Thiel *et al*, 2009; Cagnol & Chambard, 2010). Induction of mitochondrial apoptosis pathway involves Caspase-9 activation, whereas extrinsic

apoptosis pathway is triggered by tumour necrosis factors (e.g. TNF-a or FasL) binding to death domain receptors, in turn causing subsequent activation of Caspase-8 (Fulda & Debatin, 2006). Interestingly, both anti-apoptotic effects of RAF-MEK-ERK signalling (Erhardt *et al*, 1999; Lehmann *et al*, 2000; Schulze *et al*, 2001; Thiel *et al*, 2009), and pro-apoptotic effects (Wang *et al*, 2000; Zhuang & Schnellmann, 2006; Cagnol & Chambard, 2010; Martin & Pognonec, 2010; Subramaniam & Unsicker, 2010; Teixeiro & Daniels, 2010) have been reported for several cellular contexts. However, pro-apoptotic effects were more often reported in lymphocytes and cells of neuronal origin (Cagnol & Chambard, 2010). This is remarkable since HEK293 cells have been identified as of neuronal origin (Shaw *et al*, 2002).

As mentioned, HEK293ΔRAF1:ER cells in particular undergo Caspase-8-mediated apoptosis upon constant activation with 4OHT. However, the regulatory mechanism controlling Caspase-8 activity in these cells remains to be determined (Cagnol *et al*, 2006). As it was shown that Caspase-8 activation in these cells is independent of Fas-associated death domain (FADD) signalling (Cagnol *et al*, 2006), it was later speculated that the observed Caspase-8 activation might be regulated via genes from the TNF receptor super family (Cagnol & Chambard, 2010). In our data, *TNFRSF12A* has been identified as an upregulated DEG with model-derived transcriptional delay of 46 min and mRNA half-life of 183 min. Independent of its delay, its long half-life enables it to translate ERK signal duration into response amplitude. We therefore speculate mRNA upregulation of *TNFRSF12A* could account for apoptosis in HEK293ΔRAF1:ER cells exposed to prolonged ERK activation (Fig EV7B).

## Discussion

The idea of mRNA half-life being important for the kinetics of gene induction is as old as the discovery of the messenger RNA itself (Jacob & Monod, 1961). Yet only the advent of high-throughput technologies allowed to test this hypothesis in a genome-wide manner. Since that time, several studies have specifically demonstrated that short-lived transcripts respond early and that long-lived transcripts respond late to external stimuli. For example, the temporal order of gene expression was shown to be governed by mRNA half-life upon $H_2O_2$-induced stress in yeast (Shalem *et al*, 2008), upon NF-kB signalling in mouse 3T3 fibroblasts (Hao & Baltimore, 2009) and upon IL-2 signalling in murine T cells (Elkon *et al*, 2010). Using reporter constructs, it was further revealed that timing of mRNA dynamics is an intrinsic feature of the half-life encoded in the 3'UTR sequence (Hao & Baltimore, 2009).

It has been noted that protein function correlates with mRNA half-life, and in those mRNAs that need to be quickly regulated like transcription factors and other regulatory proteins tend to be short-lived (Wang *et al*, 2002; Yang *et al*, 2003; Legewie *et al*, 2008; Schwanhäusser *et al*, 2011). In agreement with these observations, we also found many transcription factors among the short-lived, fast responding mRNAs.

Interestingly, immediate–late genes (ILGs) are enriched for genes that are involved in positive regulation of apoptosis, which is the cell fate for sustained ERK signalling in our model system, suggesting that the mRNA half-life is important to functionally decode

signal duration. Very recently, the idea that mRNA half-life is involved in signal decoding has been shown for signal frequency decoding of p53 signalling (Porter *et al*, 2016). Here, short-lived transcripts relay the oscillatory pattern of p53 signalling pulses to response pulses, whereas only long-lived transcripts decode the pulses by translating them into response amplitude (Porter *et al*, 2016). In this study, we demonstrated that this principle also transfers to decoding of signal duration in ERK signalling.

Using a synthetic model system for ERK signalling combined with computational modelling of transcript kinetics, we demonstrated that mRNA longevity enables genes to translate signal duration into response amplitude. This is opposed to short-lived mRNAs that only relay signal duration to response duration. In accordance with previous research (Shalem *et al*, 2008; Hao & Baltimore, 2009; Elkon *et al*, 2010; Porter *et al*, 2016), we find that mRNA half-life is the dominating feature of expression kinetics for different input stimuli. Among primary response genes, we introduced a new cluster of immediate–late genes (ILGs) that makes use of this principle to decode ERK signal duration. In particular, response amplitude of ILGs precisely reflects ERK signal duration, as opposed to immediate–early genes (IEGs) that show no difference in response amplitude. While ILGs share similar promoter characteristics with IEGs and are also immediately induced, they differ by their mRNA half-life. We found this principle to be conserved in two different model systems for ERK signal duration (PC12 and MCF7), where long-lived genes dominate decoding of signal duration to response amplitude.

It is somewhat surprising that a simple model of gene expression that combines transcription and RNA processing into a single step of mRNA synthesis describes the data with reasonable accuracy (when complemented with a delay parameter). Over the last years, several mathematical frameworks with varying degree of complexity have been presented to estimate the contribution of processing, transcription and degradation rates from measured RNA dynamics (Zeisel *et al*, 2011; Rabani *et al*, 2014; de Pretis *et al*, 2015; Cheng *et al*, 2017). Two main conclusions arise from these studies. First, ordinary differential equations with only few parameters can accurately reflect mRNA dynamics. Secondly, changes in mRNA expression are mainly governed by changes in mRNA transcription, whereas processing and degradation rates are only altered for a minority of regulated genes (4 and 10% of genes, respectively) (Rabani *et al*, 2014). By the extension of a basic model of gene expression with a simple delay parameter, we were able to quantitatively dissect gene expression dynamics for sustained signalling where peak expression cannot be defined. Thereby, we identified temporal subclasses of PRGs (IEGs, DEGs and ILGs) with distinct functions. However, the analysis also shows that the gene expression parameters are continuous (Fig 3D and E), and therefore decode duration to varying extend. It is hence of great importance to view gene cluster definitions as a heuristic to aid interpretation and to ease comparison of results across literature.

In our study, we used a synthetic model system that mimics the activation of the oncogene RAF. Oncogenic hyperactivation of RAF1/BRAF is a pro-survival signal in many contexts. However, many cell types activate fall-back programmes to oppose the overactive signalling of RAF. Like DEGs, ILGs may serve as such a fall-back module to counteract pro-survival signals sent out by

sustained RAF activation. When benign tumours progress into malignant ones, many negative feedback mechanisms that conferred robustness before are lost (Friday *et al*, 2008). A deep understanding of feedback modules or fail-safe mechanisms in the cluster of ILGs that decode sustained oncogenic signalling is therefore crucial to better understand what distinguishes proto-oncogenic from oncogenic signalling.

# Materials and Methods

### Cell culture, microarray hybridisation and phosphoprotein assay

HEK293ΔRAF1:ER cells (Samuels *et al*, 1993; reviewed in McMahon, 2001) were cultured in complete DMEM high glucose without phenol red with 10% foetal calf serum supplemented with antibiotics (pen/strep). Before stimulation, cells were starved in serum-free medium overnight. Cells were stimulated with 4-hydroxy tamoxifen (Sigma-Aldrich H7904; 0.5 μM), U0126 (20 μM), EGF (25 ng/ml), FGF1 (50 ng/ml) or IGF (100 ng/ml). Translation was inhibited with cycloheximide (10 μM), and transcription was inhibited with actinomycin D (5 μM). RNA for microarray hybridisation was isolated with TRIzol® reagent. cDNA was fragmented, labelled and hybridised to Affymetrix Human Gene 1.0 ST Arrays. Phosphoprotein levels were assessed with Bio-Plex® (Bio-Rad) as described previously (Klinger *et al*, 2013). Metabolic labelling of RNA with 200 μM 4SU 1 hour before harvesting followed by RNA-Seq was performed as described previously (Schueler *et al*, 2014).

### Identification of differentially expressed genes from microarray data

Fluorescence intensities from scanned microarrays were processed and analysed in R. Background correction, quantile normalisation, probe set summarisation and $\log_2$ transformation were performed with help of robust multichip average algorithm (RMA) (Irizarry, 2003). Probe sets were annotated with R package hugene10sttranscriptcluster.db. All probe sets mapping to a HUGO symbol identifier were considered. For transcripts represented by multiple probe sets, the probe set with highest mean expression across samples was considered. Transcripts expressed below median expression in all samples were excluded from analysis. $\log_2$ fold changes were calculated with respect to mean expression in untreated samples (UT_1 and UT_3). UT_2 was excluded due to strong dissimilarity to UT_1 and UT_3 in cluster analysis of correlation values and putative contamination. $\log_2$ fold changes for independently obtained EGF, and FGF time course data were calculated with respect to mean expression in corresponding untreated samples (UT_1_n, UT_2_n, UT_3_n). To account for expression level-dependent variations, an empirical null model was based on replicates for 2-h 4OHT treatment. For this, transcripts were ranked by their mean expression across replicates and a moving average with window size $k = 2,000$ was calculated to serve as an expected variance measure for a given expression level. Z-scores for each transcript $p_i$ in each sample $j$ were calculated accordingly:

$$z_{i,j} = \frac{p_{i,j} - p_{i,UT}}{\left\langle sd\left(\frac{p_{i,j} - p_{i,UT}}{2}\right)\right\rangle}$$

Genes exceeding an absolute *z*-score of 5.6 in 4OHT time course data were considered regulated (1,490 upregulated, 2,037 downregulated). This corresponded to an average false discovery rate (FDR) of 1% in 4OHT time course data. Here, false positives were estimated by counting transcripts detected differentially expressed between one replicate and the mean of the two other replicates of the 2-h 4OHT treatment samples. For all downstream analyses, 4OHT-regulated genes were further filtered in two steps. First, regulated genes were tested against a random set of unregulated genes (of the same size) for their $\log_2$ fold change standard deviation ($SD_{\log2fc}$) across all samples. This was done to filter out a large fraction of erroneously detected genes, which were unaltered across all samples when the untreated condition was left out. Here, a $SD_{\log2fc}$ cut-off was defined at FDR of 5% (253 upregulated, 234 downregulated genes remained). Secondly, genes induced in a nonmonotonic fashion that could not be fitted to our one-step model were excluded from the analysis. All remaining genes (189 upregulated, 146 downregulated) are referred to as differentially expressed in the main text.

### RNA-sequencing data generation and preprocessing

Total RNA was extracted with TRIzol. Labelled and unlabelled fractions were separated as described previously (Baltz *et al*, 2012). Sequencing libraries were prepared using Illumina TruSeq mRNA Library Prep Kit v2 and sequenced on Illumina HiSeq 2000. Read files were demultiplexed, and sequencing adapters were trimmed using flexbar (Dodt *et al*, 2012). Reads were mapped with STAR aligner v2.4.1 (Dobin *et al*, 2013) on hg19 using GENCODE v19 for annotation and counted with subread featureCounts (Liao *et al*, 2014). Raw read counts were normalised with edgeR TMM (Robinson *et al*, 2010) and eventually analysed with R package DTA (Miller *et al*, 2011). The entire preprocessing pipeline was written in Snakemake (Köster & Rahmann, 2012).

### Identification of primary response genes

Differentially expressed genes were checked for significant *z*-scores in CYHX samples. Differentially expressed genes also significantly induced in any sample of parallel CYHX treatment (*z*-score > 5.6, corresponding to FDR = 1%) were considered primary response genes.

### Modelling of mRNA dynamics

Gene expression data were fitted to complete and simple model for mRNA dynamics as described in the main text using Nelder–Mead method implemented in R package optimx. For a given expression of a gene at time $t$, relative amplitude was deduced from gene-wise parameter estimates of $k_0$, $k$ and $\gamma$:

$$relative\ amplitude\,(t) = \left(expression\,(t) - \frac{k_0}{\gamma}\right) \Big/ \left(\frac{k_0 + k}{\gamma} - \frac{k_0}{\gamma}\right)$$

To obtain semi-quantitative $\log_2$ fold change predictions for growth factor-induced gene expression, gene-wise fitted model parameters and input functions for ERK-dependent transcription rate $k$ were fed to the complete model.

### Determination of mRNA half-lives based on transcription blockage with actinomycin D and metabolic labelling with 4-thiouridine

Half-life estimates based on ActD-mediated transcriptional shutdown were derived from microarray gene expression time course data (Fig EV1A). Since quantile normalisation assumes constant total RNA levels across samples (Bar-Joseph *et al*, 2012), RMA was performed without quantile normalisation for ActD samples. Samples were instead normalised to median expression of 61 long-lived mRNAs ($t_{1/2} > 16$ h) consistently identified in two published data sets on human mRNA half-life (Yang *et al*, 2003; Friedel *et al*, 2009). Both time series were than fitted to an exponential decay function of form $M(t) = M_0 + e^{-\gamma t}$ to infer decay rates $\gamma$.

Half-life estimates based on metabolic labelling with 4SU followed by RNA sequencing were calculated using all three fractions of RNA, that is total RNA, labelled RNA (eluate) and unlabelled RNA (flow-through). Dynamic transcriptome analysis (DTA) was used for quantification (Miller *et al*, 2011).

Median, mean and standard deviation of half-lives for all expressed genes were calculated from the three different data sets (ActD ON, ActD OFF, 4SU) and are provided as supplementary data (Table EV2).

### Analysis of PC12 and MCF7 data

Published time course expression raw data on PC12 (Offermann *et al*, 2016) and MCF7 (Saeki *et al*, 2009) were downloaded from Gene Expression Omnibus (accession numbers: GSE74327 and GSE13009). Data were preprocessed and analysed analogously to HEK293ΔRAF1:ER microarray data presented in this work.

### qRT–PCR primers, Western blot antibodies and flow cytometry

cDNA was synthesised using High-Capacity RNA-to-cDNA™ Kit (Applied Biosystems #4387406). qRT–PCR was performed using Taqman gene expression assay (Thermo Fisher #4304437) with following Taqman primers (Thermo Fisher): Hs01045540_g1 (ARC), Hs00156548_m1 (CLU), Hs00610256_g1 (DUSP1), Hs01044001_m1 (DUSP6), Hs00152928_m1 (EGR1), Hs00166165_m1 (EGR2), Hs00170630_m1 (FOS), Hs00171851_m1 (FOSB), Hs04187685_m1 (FOSL1), Hs00357891_s1 (JUNB), Hs00374226_m1 (NR4A1), Hs00943178_g1 (PGK1), Hs00169585_m1 (PPP1R15A), Hs00153 133_m1 (PTGS2), Hs04334126_m1 (TFPI2), Hs00959047_g1 (TNFRSF12A), Hs00381614_m1 (ZCCHC12), Hs00185658_m1 (ZFP36).

Protein was extracted using Bio-Rad Cell Lysis Buffer (#171-304006M). Concentration was determined using Thermo Fisher Pierce BCA Protein Assay (#23228). 25–50 μg of purified protein was used for blotting. Images were acquired using Li-Cor Odyssey Scanner. Western blot antibodies were as follows: EGR1 (Santa Cruz sc-110), FOS (Cell Signaling #2250), CLU (Santa Cruz sc-8354), FOSL1 (Santa Cruz sc-376148).

For flow cytometry, cells were harvested 48 h after treatment and fixed in 2% paraformaldehyde (PFA) for 10 min at RT. Cells were permeabilised in methanol and incubated on ice for 30 min. For immunostaining, cells were incubated for 1 h with Cleaved Caspase-3 rabbit mAb (Cell Signaling #9602).

### Data availability

Both microarray gene expression data and metabolic labelling RNA-Seq data are accessible from gene expression omnibus (GEO) under SuperSeries accession number GSE93611.

**Expanded View** for this article is available online.

### Acknowledgements
We thank S. Cagnol and P. Lenormand for providing the HEK293ΔRAF1:ER cell line and N. Lehmann and C. Stein for technical assistance. Funding was provided by Berlin School of Integrative Oncology (to FU) and Berlin Institute of Health (to NB and ML) and German Ministry of Education and Research (BMBF, grant MapTorNet to NB).

### Author contributions
FU performed computational analyses. AS, EW, FU and RF-G conducted experiments. FU and NB wrote the manuscript with input from AS, EW, RF-G, JM, ML and BK. FU, BK and NB designed the experiments.

### Conflict of interest
The authors declare that they have no conflict of interest.

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
