## [Review Process File · Molecular Systems Biology]

An immediate-late gene expression module decodes ERK signal duration

Mr. Florian Uhlig, Mrs. Anja Sieber, Emanuel Wyler, Raphaela Fritsche-Guenther, Johannes Meisig, Markus Landthaler, Bertram Klinger and Nils Blüthgen

Corresponding author: Nils Blüthgen, Charité Universitätsmedizin Berlin

Review timeline:

Original Submission date:	8 November 2015
Initial Decision:	4 December 2015
Resubmission date:	26 January 2017
Editorial Decision:	05 March 2017
Revision received:	29 March 2017
Accepted:	04 April 2017

Editor: Maria Polychronidou

Transaction Report:

Initial Editorial Decision

04 December 2015

Thank you again for submitting your work to Molecular Systems Biology. We have now heard back from the three referees who were asked to evaluate your manuscript. As you will see from the reports below, the referees raise substantial concerns on your work, which, I am afraid to say, preclude its publication in Molecular Systems Biology.

The referees acknowledge that the topic is timely and that the experimental set-up is interesting. However, they point out that as it stands the study remains preliminary. In particular, they think that it is not convincingly demonstrated which group of genes (i.e. the immediate long-lived genes (ILGs)) is the one that distinguishes between transient and sustained ERK signaling, while the biological relevance of the ILGs in determining cell fate decisions remains unclear.

We have also circulated the reports to all reviewers as part of our 'pre-decision cross-commenting' policy. During this process, reviewer #2, mentioned:

"I agree with all the points raised regarding deficiencies of data presentation and analysis. However, the two deficiencies that go to the scope of the study are also important and would precede my criticism of the data analysis/presentation:

1. The correlation between mRNA half-life and responsiveness to ERK activities of different duration was not substantiated with a functional demonstration in experiments: swap 3'UTRs of mRNAs and promoters to show that the responsiveness really is due to the half-life control not promoter inducibility.

2. We would like to know something about the biological relevance of this regulatory principle: Does it apply to mRNAs induced in PC12 cells by NGF and EGF? If so which? Does it apply to their resulting proteins (Western blots)? Do these proteins likely play a role in the cell fate decision making?

(I don't think that it is not necessary within the scope of this paper to demonstrate by knockout (requirement) or overexpression (sufficiency) that these proteins really do mediate the cell fate decision process of proliferation vs differentiation.)

Overall, the reviewers rated the validity of the main conclusions as "medium/low" and indicated that, in its current form, the work is not suitable for publication in *Molecular Systems Biology*. Considering that our journal policy allows in principle a single round of major revision and that the outcome of the additional experimentation and analyses suggested by the reviewers is unclear, we see no other choice but to reject the manuscript.

Nevertheless, in view of the importance of the topic addressed and considering that the reviewers did have positive words for the goals of the study and appreciated the elegant experimental set-up, we would be willing to consider a new and extended manuscript based on this work, provided that the issues raised by the reviewers are convincingly addressed. We recognize that this would involve substantial additional experimentation and analyses with unclear outcome and, as you probably understand, we can give no guarantee about its eventual acceptability.

If you do decide to follow this course then it would be helpful to enclose with your re-submission an account of how the work has been altered in response to the points raised in the present review.

REFEREE REPORTS

Reviewer #1:

The paper from the group of Nils Blüthgen identifies a new class (ILG) of primary response genes (PRG) based on gene expression kinetics and the subsequent half-lifetime determination. The authors claim that the long-lived ILGs can decode the difference between sustained and transient ERK response profiles. The basic idea to use synthetic stimuli to control ERK response and measure the subsequent temporal gene expression profiles is excellent. However there seems to be several layers of problems ranging from data acquisition, analysis and logic behind the interpretation. There is no evidence in this paper that the cell can read out the maximum relative induction of ILGs as a way to distinguish ERK responses. In fact, Fig.2 shows that IEGs would be the most sensitive way to distinguish between sustained and transient response. The rapid decay of the IEGs maximizes the difference in their response profiles to the two stimuli. In contrast, this difference is marginal and would be very sensitive to noise in the case of ILGs. In both cases, this difference happens at the longer time scale after ERK response (Fig. 2B). What is a real pity is that the authors developed a nice protocol to synthetically vary ERK pulse length but did not study its effects on gene expression kinetics.

Specific points:

1. The authors have not specified in the manuscript the number of repetitions for each gene expression experiment. This makes it difficult to judge the robustness of the data and does not allow the uncertainty of the fitted parameters to be determined. This is essential for reliable conclusions in the paper, especially when comparing gene expression profiles in response to different stimuli.
2. In Fig. 1, why are the gene expression profiles after CHX treatment not shown on a 10h time scale, but rather on a 4h scale? This is essential to robustly separate PRGs from SRGs that are expected to kick in later.
3. Additionally, the color bars throughout the Figures are not helpful to discriminate the differences between the response profiles. In essence, a 8 bit dynamic range is represented by 2 bits and most of the relevant positive range is covered by only one red color. This is apparent in Fig.1 and 5 and to a lesser extent in the gray scale of Fig.3.

4. In Fig.2b, it is clear that the biggest difference between a sustained and transient ERK response occurs in IEGs and not ILGs at later time points, demonstrating that IEGs should be the most sensitive in distinguishing between the two ERK responses. It is not clear that the difference in the maximal amplitude of the ILGs is a robust readout of ERK response. Especially, due to the slow ILG's relaxation time, the signal is insensitive to differences in ERK relaxation response.
5. In Fig.3C, the authors have not quantified the comparison between the model and the data to be able to judge the goodness of fit. There is also no variance on the fitted parameters due to lack of experiment repetition. The comparison between the extended and the constrained model is in principle valid, however it is less meaningful without the variance in the data. Additionally, the authors introduce an arbitrary threshold of $\Delta t < 30\text{min}$ to represent data that could only be fitted by the extended model as if there is no delay. This is a completely artificial way of excluding DEGs.
6. Why in Fig. 3 did the authors not perform this experiment with a synthetic pulse to obtain good parameter values for the mRNA decay rates? This is especially important since later on they claim that ERK response affects this parameter (Fig.4F).
7. In Fig.5 again the authors do not quantitatively compare the model and the experimental data. At least residual plots should be shown. The difference in the experimental temporal profiles upon EGF and FGF stimulus seem marginal, especially in the ILG profiles. This is again the problem of data representation as discussed in point 3. I also do not see the relevance of the IGF data.

Reviewer #2:

Uhlitz et al address how the temporal modulation of ERK activity controls gene expression. Their study is motivated by the classic studies of PC12 cell activation in response to NGF and EGF where the duration of ERK was shown to be important for cellular decision making. However, they use here an artificial, engineered system that has the advantage of controlling ERK activity but the disadvantage of being more removed from the biological phenomenon. In their studies, they identify mRNA half-life as controlling the expression of one cohort of genes.

Introduction states that "'How can primary genes respond late?' Is an important question." Yet, several studies, including Hao et al 2009 shows that mRNA half-life controls the timing of gene expression in response to TNF.

Still, to my knowledge it is that whether long duration signaling is required for the activation of long half-life mRNAs has not been established in any experimental system.

The authors do demonstrate this, though the impact is reduced as the experimental system is artificial, and they make no attempt to show that the genes that they identify to be under the control of this dynamic coding, in fact play a role in the biology of ERK-dynamics-mediated cell fate decision-making. Thus the work remains a proof of principle experimental study of the widely-appreciated theoretical prediction that a long molecular half-life can function as a signal duration decoder. It would be so much more rewarding to know that this mechanism had biological relevance in determining cell fate decisions.

Some specific points (not exhaustive, given preceding comments)

"Long-lived genes" is a poor phrase. The authors refer to mRNA. These are long-lived mRNAs.

Why is ΔT part of the mRNA degradation term? "Biologically, the additional delay parameter Δt here accounts for all steps that need to take place before transcription can start, like chromatin remodelling, transcription factor recruitment, and polymerase recruitment." This does not rationalize it.

Transcriptional delays can have a profound effect in decoding transcription factor dynamics (Hao et al papers). These previous papers present an elegant modeling solution but the present study does not consider TFs, only the upstream kinase.

Model fitting to data is not very sophisticated. Data errors from replicates do not seem to be considered. Chi squared tests do not consider dynamic profiles only datapoints.

No attempt is made to address protein half-life and the biological or regulatory consequences of that.

There are many mRNAs for which this regulatory mechanism does not hold, but no attempt is made to provide an explanation for those.

Reviewer #3:

Uhlitz et al. present a quantitative study of the kinetics of ERK-induced gene transcription, in which they identify a class of genes, the ILGs, with induction kinetics distinct from the previously described immediate early genes and delayed early genes. They demonstrate using a combination of experiments and modeling that these genes are induced immediately as part of the primary response but have longer mRNA half-lives, resulting in a slower apparent response. They differentiate this form of regulation from genes that have an apparent delay in the onset of transcription (the DEGs), likely due to promoter structure. Overall, I found this study to be informative and thought-provoking. The authors add some important details to the model of MAPK-driven gene expression, explaining some apparent contradictions and resulting in a better overall model for this central and well-studied process. The experimental data appears technically sound, and should be a useful resource for others in the field.

My main criticisms deal with the presentation of the data, which should be strengthened at a number of points.

Major comments

1. A strength of this paper is what appear to be the very nice datasets from which the models are derived. However, it is unfortunate that the gene-by-gene data have not been included as supplementary data. In particular, the half-life analyses are only presented in graphical forms that make it impossible to glean much specific information. Including the data (ideally non-normalized) would significantly improve the value of this paper as a resource, and also make it easier to judge the strength of the data in supporting the conclusions.
2. Some important experimental details have been left out. For example, the time points analyzed in the ActD half life experiment are not listed, making it impossible to judge the time resolution of the assay. The presentation of Fig. 4D is also quite difficult to understand. Please provide details on how the relative gene counts and RNA fractions are calculated.
3. Discrimination of IEGs vs. ILGs is presented as a function of the mRNA half-life and much of the manuscript discusses genes and their functions in terms of these clusters. However, it seems that the arbitrary classification guidelines used in the past and here may not be ideal to describe the distributions of genes and functions. The mRNA half-lives and transcriptional delays reported do not appear to be discretely binned, but rather continuously distributed over the genes observed. Similarly, the GO terms reported show some enrichments based on the classifications, but not complete association of particular functions with IEGs, DEGs, or ILGs. This study instead seems to provide evidence that these classifications may be obsolete. The continuous nature of varied mRNA stability and transcriptional delay should be discussed, ideally offering a clear view of how gene responses are distributed.
4. All of the data and model outputs considered are normalized to a maximal expression level (i.e. at steady state under a maximally inducing signal). It is important to consider that when viewing normalized outputs ('percent induction' or 'relative induction', here), quantitative aspects are obscured, which may be important to the effect of the signal. For example, if the only difference between the inductions of two genes is mRNA half-life, the gene with a longer half-life will appear more weakly expressed in the short term when viewed as percent induction (as in Fig. 2A). However, the actual concentration of that gene will rise faster and to higher levels than for a gene with a shorter half-life.

Minor comments

1. Fig. 5C presents corroborating evidence on prediction based on the parameterized model, but without any quantification of the prediction error. Please provide a metric of prediction error (e.g. mean squared error) on a gene by gene basis so the distribution can be inspected.

2. The response of the IEGs to different durations of ERK activity has been the subject of much study, albeit at the protein level, not mRNA. It would be useful to compare this study's results to these earlier studies, ideally providing some thoughts on how the mRNA and protein level regulation may be integrated.

Resubmission

26 January 2017

Response to Editorial comments

In particular, they think that it is not convincingly demonstrated which group of genes (i.e. the immediate long-lived genes (ILGs)) is the one that distinguishes between transient and sustained ERK signaling, while the biological relevance of the ILGs in determining cell fate decisions remains unclear.

We agree that IEGs and DEGs also behave differently between transient and sustained signalling. However, they differ primarily in their response duration, and not in their response amplitude. That means they don't *decode* the signal, but they *relay* the signal and the decoding is *postponed* (e.g. to the protein level, such as in case of FOS protein stabilisation, cf. new Fig 6E and lines 314-320). For ILGs, signal duration is instead translated into response *amplitude*. We generated now extensive data that clearly shows that ILGs convert duration into amplitude, while IEGs do not (cf. Fig 2 + lines 148-151, as well as our new data on different signal durations in Fig 6C-E, Fig EV6 A-B + lines 301-320).

We now also show that the principle of signal duration being translated into response amplitude is conserved in rat PC12 cells and human MCF7 cells, two cell systems which serve as paradigm models for cell fate decisions based on signal duration (cf. new Fig 6F, and Fig EV6 C-D + lines 321-338).

We have also circulated the reports to all reviewers as part of our 'pre-decision cross-commenting' policy. During this process, reviewer #2, mentioned:

"I agree with all the points raised regarding deficiencies of data presentation and analysis. However, the two deficiencies that go to the scope of the study are also important and would precede my criticism of the data analysis/presentation:

1. The correlation between mRNA half-life and responsiveness to ERK activities of different duration was not substantiated with a functional demonstration in experiments: swap 3'UTRs of mRNAs and promoters to show that the responsiveness really is due to the half-life control not promoter inducibility.

We agree that synthetically altering 3'UTRs to modulate mRNA half life would be an interesting approach, but this will also alter steady-state expression, which then makes interpretation difficult. Instead, we decided to use metabolic pulse labelling to measure nascent RNA. This data provides evidence that promoter inducibility is similar between IEGs and ILGs, but delayed for DEGs. Hence, mRNA half-life is the feature that determines lateness of ILGs. This can also be seen in PC12 and MCF7 cells, where mRNA half-life is a strong predictor for response time. We added both metabolic pulse labelling data and comparison to PC12/MCF7 to the manuscript (Fig 5C, 6F, EV5, EV6 C-D, cf. lines 255-269 + 321-338).

2. We would like to know something about the biological relevance of this regulatory principle: Does it apply to mRNAs induced in PC12 cells by NGF and EGF? If so which?

This is an excellent idea, and we analysed gene expression time course data of PC12 cells and also included MCF7 cells, another paradigm model system for cell fate decisions based on ERK signal duration. We find a high correlation of gene response times in our model system with peak expression times in PC12 and MCF7 cells, showing that the overall principle applies also to these more relevant cell systems (Fig EV6 C-D). Moreover, we show that in these cell systems, decoding

of duration to amplitude is clearly governed by ILGs. As mentioned above, this data is shown in Fig. 6F and described in lines 321-338.

Does it apply to their resulting proteins (Western blots)

This is an important point. Clearly, IEG with long protein half life could similarly decode duration. We decided to focus on the protein expression of four proteins, where good antibodies are available and which represent ILGs and IEGs: FOS (an example of IEG with a protein that is stabilized by the signal), EGR1 (short lived protein IEG), FOSL1 and CLU (ILGs). The data on the IEGs suggest that long lived proteins can also decode signal duration, and it clearly shows that decoding of signal duration by ILGs is propagated to the protein level. This is now shown in Fig 6E and EV6 B and described in lines 314-320.

Do these proteins likely play a role in the cell fate decision making? (I don't think that it is not necessary within the scope of this paper to demonstrate by knockout (requirement) or overexpression (sufficiency) that these proteins really do mediate the cell fate decision process of proliferation vs differentiation.)

We agree with the referee that extensive functional studies are beyond the scope of the study. However, HEK293ΔRAF1:ER cells clearly differentiate short and prolonged ERK activity. Prolonged ERK activity triggers apoptosis, whereas short activity does not (as we now demonstrate in new Fig EV7 B). ILGs are enriched for Gene Ontology term *positive regulation of apoptotic process* (Fig EV7 A).

Response to referee reports

Referee 1

The paper from the group of Nils Blüthgen identifies a new class (ILG) of primary response genes (PRG) based on gene expression kinetics and the subsequent half-lifetime determination. The authors claim that the long-lived ILGs can decode the difference between sustained and transient ERK response profiles. The basic idea to use synthetic stimuli to control ERK response and measure the subsequent temporal gene expression profiles is excellent. However there seems to be several layers of problems ranging from data acquisition, analysis and logic behind the interpretation.

We thank the referee for these supporting comments and constructive criticism, and have addressed the issues raised as detailed in the following.

There is no evidence in this paper that the cell can read out the maximum relative induction of ILGs as a way to distinguish ERK responses. In fact, Fig.2 shows that IEGs would be the most sensitive way to distinguish between sustained and transient response. The rapid decay of the IEGs maximizes the difference in their response profiles to the two stimuli. In contrast, this difference is marginal and would be very sensitive to noise in the case of ILGs. In both cases, this difference happens at the longer time scale after ERK response (Fig. 2B). What is a real pity is that the authors developed a nice protocol to synthetically vary ERK pulse length but did not study its effects on gene expression kinetics.

We agree with the referee that there is a strong difference of IEGs at later times for short vs. long input, i.e. IEGs respond to short stimuli with short expression, and to long stimuli with long lasting expression. Therefore, IEGs do not *decode* signal duration, they *relay* signal duration, and thus *postpone* the problem (we now stress this difference throughout the manuscript, cf. lines 59-62, 272-276, 286-288, 394-396). In response to this comment and further comments, we also generated extensive q-RT-PCR data for various input pulse lengths that clearly show that IEGs relay signal duration, whereas ILG and other long-lived DEGs decode signal duration into amplitude (cf. new

Fig 6C, 6D, EV6 A, + lines 301-313). Using expression time series data from PC12 and MCF7 cells, we show that this principle is conserved in these cell types (cf. new Fig 6F, EV6 C-D + lines 321-338).

With respect to noise characteristics, we can only speculate without data on the single cell level, which would be beyond the scope of this manuscript. However, theoretically long-lived mRNAs filter high frequency noise, and short-lived genes should be more sensitive to this. Therefore, depending on noise characteristics, ILGs may be even less noise sensitive.

Specific points:

1. *The authors have not specified in the manuscript the number of repetitions for each gene expression experiment. This makes it difficult to judge the robustness of the data and does not allow the uncertainty of the fitted parameters to be determined. This is essential for reliable conclusions in the paper, especially when comparing gene expression profiles in response to different stimuli.*

All acquired samples are now visualized in Fig EV1. Both microarray gene expression data and metabolic labelling RNA-Seq data is accessible from gene expression omnibus (GEO) under SuperSeries accession number GSE93611. Reviewer access link:

<https://www.ncbi.nlm.nih.gov/geo/query/acc.cgi?token=ufancqisjtqhdiz&acc=GSE93611>

2. *In Fig. 1, why are the gene expression profiles after CHX treatment not shown on a 10h time scale, but rather on a 4h scale? This is essential to robustly separate PRGs from SRGs that are expected to kick in later.*

CYHX is a severe cellular toxicant, precluding it from longer treatments. All genes significantly regulated upon 1, 2 or 4h of CYHX+4OHT were considered PRGs. Hence, our PRG identification has high precision but might lack sensitivity for very late induced PRGs. Correspondingly, SRG identification has high sensitivity but might lack precision. Since we focus our analysis on IEGs and ILGs, we think that it is important to have high precision for the PRGs. PRG identification is further backed by our new metabolic labelling data that shows the immediate transcription of IEGs and ILGs (cf. new Fig 5B and EV5 B).

3. *Additionally, the color bars throughout the Figures are not helpful to discriminate the differences between the response profiles. In essence, a 8 bit dynamic range is represented by 2 bits and most of the relevant positive range is covered by only one red color. This is apparent in Fig.1 and 5 and to a lesser extent in the gray scale of Fig.3.*

We agree that the color code was suboptimal and updated it accordingly.

4. *In Fig.2b, it is clear that the biggest difference between a sustained and transient ERK response occurs in IEGs and not ILGs at later time points, demonstrating that IEGs should be the most sensitive in distinguishing between the two ERK responses. It is not clear that the difference in the maximal amplitude of the ILGs is a robust readout of ERK response. Especially, due to the slow ILG's relaxation time, the signal is insensitive to differences in ERK relaxation response.*

We agree that IEGs are the most sensitive to translate signal *duration* to response *duration*. However, only long-lived genes such as ILGs and other long-lived DEGs are capable of translating signal *duration* to response *amplitude*. This is now clearly demonstrated in the updated manuscript using extensive q-RT-PCR analysis and analysis of PC12 and MCF7 data (see above).

5. *In Fig.3C, the authors have not quantified the comparison between the model and the data to be able to judge the goodness of fit. There is also no variance on the fitted parameters due to lack of experiment repetition. The comparison between the extended and the constrained model is in principle valid, however it is less meaningful without the variance in the data. Additionally, the authors introduce an arbitrary threshold of $\Delta t < 30\text{min}$ to represent data that could only be fitted by the extended*

model as if there is no delay. This is a completely artificial way of excluding DEGs.

We agree that quantitative evaluation of goodness of fit and goodness of predictions are important to evaluate the model. We now present mean error values to quantify goodness of fit (cf. updated Fig 3C) and goodness of gene expression prediction (cf. updated Fig 4C + lines 209-213) alongside with the model fits and model predictions, respectively.

Clearly, model selection requires estimates of data variance. In accordance with the state-of-the-art model selection literature, we used variance estimates based on an error model trained on replicate arrays (relating variance and mean expression), and a likelihood ratio test based on chi-square test statistics (Fig EV2 + lines 166-171). The threshold of $\Delta t < 30\text{min}$ is a consequence of sampling precision in our measurements. An attempt to fit smaller delays results in non-identifiable partitioning between delay parameter and half-life. Most importantly, the main point of our paper (decoding by mRNA half-lives) is valid for ILGs and long-lived DEGs, i.e. this particular discrimination is not crucial for our analysis.

6. *Why in Fig. 3 did the authors not perform this experiment with a synthetic pulse to obtain good parameter values for the mRNA decay rates? This is especially important since later on they claim that ERK response affects this parameter (Fig.4F).*

We cannot agree more that metabolic labeling is better suited to derive half-lives. However, deriving half-lives using metabolic labeling requires steady-state conditions, and we are particularly interested in those genes that change strongly. We calculated half-lives using metabolic labeling for unstimulated cells, and show that pre-treatment half-life derived from ActD correlate well with those derived from metabolic labeling (Fig. EV4 B + lines 216-240). However, we also noticed that half-lives change when the cells are stimulated (Fig EV4 A), and therefore we decided to stick to ActD-derived half-lives. A systematic comparison of the derived half-lives (also with literature values) can be found in Figure 5A and EV4.

7. *In Fig.5 again the authors do not quantitatively compare the model and the experimental data. At least residual plots should be shown. The difference in the experimental temporal profiles upon EGF and FGF stimulus seem marginal, especially in the ILG profiles. This is again the problem of data representation as discussed in point 3. I also do not see the relevance of the IGF data.*

We added quantification of prediction error (see above), and changed the color-code accordingly. We agree that the data on IGF treatment does not add value to the manuscript and removed this data.

Referee 2:

Uhlitz et al address how the temporal modulation of ERK activity controls gene expression. Their study is motivated by the classic studies of PC12 cell activation in response to NGF and EGF where the duration of ERK was shown to be important for cellular decision making. However, they use here an artificial, engineered system that has the advantage of controlling ERK activity but the disadvantage of being more removed from the biological phenomenon. In their studies, they identify mRNA half-life as controlling the expression of one cohort of genes.

Introduction states that "How can primary genes respond late?" Is an important question." Yet, several studies, including Hao et al 2009 shows that mRNA half-life controls the timing of gene expression in response to TNF.

Still, to my knowledge it is that whether long duration signaling is required for the activation of long half-life mRNAs has not been established in any experimental system.

We thank referee 2 for the supporting comments, and the constructive criticism which helped us to improve the manuscript.

The authors do demonstrate this, though the impact is reduced as the experimental system is artificial, and they make no attempt to show that the genes that they identify to be under the control of this dynamic coding, in fact play a role in the biology of ERK-dynamics-mediated cell fate decision-making. Thus the work remains a proof of principle experimental study of the widely-appreciated theoretical prediction that a long molecular half-life can function as a signal duration decoder. It would be so much more rewarding to know that this mechanism had biological relevance in determining cell fate decisions.

We agree that this is important, and based on this comment and the comments of referee 1 we now added a comparison to two other cell systems which are considered paradigms of cell fate decisions. We show that both timing of peak expression and duration decoding are conserved also in PC12 and MCF7 cells (cf. new Fig 6F and EV6 C-D + lines 321-338).

Some specific points (not exhaustive, given preceding comments)

"Long-lived genes" is a poor phrase. The authors refer to mRNA. These are long-lived mRNAs.

We fully agree. We therefore decided to name these genes immediate late genes (ILGs) instead.

Why is delta T part of the mRNA degradation term? "Biologically, the additional delay parameter Δt here accounts for all steps that need to take place before transcription can start, like chromatin remodelling, transcription factor recruitment, and polymerase recruitment." This does not rationalize it.

Thank you for pointing this out. This was a mistake in the equation in the manuscript, and not implemented as such. We corrected the mistake.

Transcriptional delays can have a profound effect in decoding transcription factor dynamics (Hao et al papers). These previous papers present an elegant modeling solution but the present study does not consider TFs, only the upstream kinase.

We agree that delays in transcription factor activation are important, and could e.g. be captured by latent variable models. We are not sure exactly which paper the referee refers to. The Hao & Baltimore papers do not contain models, and they consider much shorter timescales than our paper, where the precise transcription initiation events and splicing rates are important. However, we consider ERK activity in our study as more reliable and also relevant readout, as this is the signal that the cells need to decode. We model delays in initiation using one explicit delay term. The newly included metabolic pulse labeling data shows that within our resolution, genes are induced within the first hour, and roughly follow the ERK kinetics (cf. new Fig 5C + lines 255-269).

Model fitting to data is not very sophisticated. Data errors from replicates do not seem to be considered. Chi squared tests do not consider dynamic profiles only datapoints.

This is not correct. We use a maximum likelihood approach, which of course considers the errors from replicates (using an error model, as detailed in the materials and methods part, which is more appropriate than "calculating" errors directly from replicates as variance estimators perform poorly for small n).

To compare models, we used the likelihood ratio test (which is based on the chi-square distribution), which is the appropriate test to compare likelihoods of two nested models (see e.g. Kreutz and Timmer 2009). We clarified the relevant passages (cf. lines 166-171)

No attempt is made to address protein half-life and the biological or regulatory consequences of that.

We agree that this is an important point and we included this in the revised version. We show now protein expression dynamics with western blot. We used also the IEG FOS, where the protein is stabilized and which is an excellent example where long-lived proteins decode duration (cf. Fig 6E, EV6 B + lines 314-320).

There are many mRNAs for which this regulatory mechanism does not hold, but no attempt is made to provide an explanation for those.

While clearly not all mRNAs are regulated in this way, it is surprising that the simple model reflects the dynamics of the vast majority of all primary response genes induced by ERK signalling (cf. Fig 4).

Reviewer 3:

Uhlitz et al. present a quantitative study of the kinetics of ERK-induced gene transcription, in which they identify a class of genes, the ILGs, with induction kinetics distinct from the previously described immediate early genes and delayed early genes. They demonstrate using a combination of experiments and modeling that these genes are induced immediately as part of the primary response but have longer mRNA half-lives, resulting in a slower apparent response. They differentiate this form of regulation from genes that have an apparent delay in the onset of transcription (the DEGs), likely due to promoter structure. Overall, I found this study to be informative and thought-provoking. The authors add some important details to the model of MAPK-driven gene expression, explaining some apparent contradictions and resulting in a better overall model for this central and well-studied process. The experimental data appears technically sound, and should be a useful resource for others in the field.

My main criticisms deal with the presentation of the data, which should be strengthened at a number of points.

We thank the referee for the support and have fundamentally changed the manuscript to improve the presentation of data as detailed below.

Major comments

1. A strength of this paper is what appear to be the very nice datasets from which the models are derived. However, it is unfortunate that the gene-by-gene data have not been included as supplementary data. In particular, the half-life analyses are only presented in graphical forms that make it impossible to glean much specific information. Including the data (ideally non-normalized) would significantly improve the value of this paper as a resource, and also make it easier to judge the strength of the data in supporting the conclusions.

We agree that this is a valuable resource, and added extensive supplementary data tables, which also includes extensive data generated during revision (cf. Table EV1+EV2). Model parameters are provided for all induced genes and mRNA half-life values for all expressed genes. Non-normalized raw data from microarrays and RNA-Seq can be accessed at GEO (see above).

2. Some important experimental details have been left out. For example, the time points analyzed in the ActD half life experiment are not listed, making it impossible to judge the time resolution of the assay. The presentation of Fig. 4D is also quite difficult to understand. Please provide details on how the relative gene counts and RNA fractions are calculated.

We have now added a Figure (Fig EV1) that displays all samples that are used in the manuscript. We updated the materials and methods section to fully describe data processing of RNA-sequencing (cf. lines 446-457).

3. Discrimination of IEGs vs. ILGs is presented as a function of the mRNA half-life and much of the manuscript discusses genes and their functions in terms of these clusters. However, it seems that the arbitrary classification guidelines used in the past and here may not be ideal to describe the distributions of genes and functions. The mRNA half-lives and transcriptional delays reported do not appear to be discretely binned, but rather continuously

distributed over the genes observed. Similarly, the GO terms reported show some enrichments based on the classifications, but not complete association of particular functions with IEGs, DEGs, or ILGs. This study instead seems to provide evidence that these classifications may be obsolete. The continuous nature of varied mRNA stability and transcriptional delay should be discussed, ideally offering a clear view of how gene responses are distributed.

We cannot agree more that mRNA stability and transcriptional delay are continuous parameters, and we show that many derived properties (decoding, timing) are quantitatively depending on mRNA stability and transcriptional delay. Simulations in figure 2B and fitted parameters in figure 3E intend to visualise the continuous nature and timing of mRNA response (also cf. lines 182-184). However, we feel that putting the genes into clusters is a heuristic that aids interpretation also in the light of previous literature. We now included this reflection in the revised manuscript (cf. lines 191-194).

4. All of the data and model outputs considered are normalized to a maximal expression level (i.e. at steady state under a maximally inducing signal). It is important to consider that when viewing normalized outputs ('percent induction' or 'relative induction', here), quantitative aspects are obscured, which may be important to the effect of the signal. For example, if the only difference between the inductions of two genes is mRNA half-life, the gene with a longer half-life will appear more weakly expressed in the short term when viewed as percent induction (as in Fig. 2A). However, the actual concentration of that gene will rise faster and to higher levels than for a gene with a shorter half-life.

This is absolutely correct and we included this important remark to guide interpretation of our results (lines 141-147): It is important to note that response amplitudes are presented as relative values normalised to steady-state expression. Such normalised values ease the comparison of the timing between different genes during their transition from one steady-state to another. At the same time however, this representation cannot reflect absolute changes in mRNA concentration. Hence, we present relative changes in expression (noted as amplitude [%]) when describing the relation between mRNA half-life and signal duration decoding and absolute changes in expression (noted as \log_2 fold change), when we focus on quantitative aspects of mRNA expression. In this regard, it is also interesting to note that short half-lives are compensated by higher transcription rates (cf. new Fig EV5) to yield similar levels of total mRNA concentration. We think that this observation and the fact that we show \log_2 fold changes were appropriate could reduce worries about obscured quantitative aspects.

Minor comments

1. Fig. 5C presents corroborating evidence on prediction based on the parameterized model, but without any quantification of the prediction error. Please provide a metric of prediction error (e.g. mean squared error) on a gene by gene basis so the distribution can be inspected.

We agree that this is important and now discuss and show mean error of gene expression predictions (cf. updated Fig 4C, new Fig EV3 + lines 209-213).

2. The response of the IEGs to different durations of ERK activity has been the subject of much study, albeit at the protein level, not mRNA. It would be useful to compare this study's results to these earlier studies, ideally providing some thoughts on how the mRNA and protein level regulation may be integrated.

We include new data for FOS and FOSL1 (FRA1), two paradigm examples for duration decoding, and a discussion of FOS protein stabilization as a paradigm on the protein level (cf. new Fig 6E, new Fig EV6 B + lines 314-320).

Thank you again for submitting your work to Molecular Systems Biology. We sent your manuscript to the previous reviewers #2 and #3 (current reviewer #1). We have now heard back from them and as you will see from the comments below, they both think that the revised/extended version of the study is significantly improved.

However, as you will see below, reviewer #2 raises concerns regarding the conceptual novelty and the biological significance of the study. We have circulated the reports to both reviewers as part of our 'pre-decision cross-commenting' process, and reviewer #1 mentioned: "I agree with reviewer 2's point that it is well established in the literature that mRNA half-life governs the kinetics of expression. However, my perspective is that this manuscript's value lies in 1) demonstrating that this concept applies to the ERK pathway, which despite being one of the most well-studied gene regulatory pathways has not been studied in this regard and 2) providing a comprehensive set of data on mRNA kinetics that will be useful for others working on this pathway. Thus, I think that the concerns about novelty are not sufficient to preclude publication and can be addressed with a balanced discussion of the literature. With regard to biological significance, I think the authors have made at least a reasonable effort to investigate this in Fig. EV7. While I agree that it would be very interesting to determine which genes underlie the proliferation/differentiation switch in the classic PC12 system, it seems to me that the multiplex nature of the system would put the complexity of the experiments required out of the scope of the study. (i.e., would it be possible to knock down enough ILGs to shift the NGF response from differentiation to proliferation? Would knockdown experiments even be sufficient, or would the steady state levels of multiple genes need to be tuned experimentally?) Other papers in this vein (for example, the Porter and Batchelor paper) have also found it difficult to ascribe defined biological functions to classes of mRNAs with different kinetics. Nonetheless, if a defined set of experiments to convincingly address this point could be prescribed, I would support asking the authors to attempt it."

Taken together, we think that additional experiments are not mandatory for the acceptance of the manuscript. Of course, we would not be opposed to the inclusion of such analyses if you have already performed them or feel inclined to do so. In line with the comment of reviewer #1, we would ask you to include a balanced discussion of previous studies on this topic. Moreover, reviewer #1 raises concerns regarding minor mistakes in the text, which should be addressed in a revision.

REFEREE REPORTS

Reviewer #1:

The authors have addressed all of our concerns adequately, and the added data on mRNA half-life strengthens the manuscript overall. However, there remain a number of confusing errors, in particular figure labeling issues for Fig 5 and Fig EV5 which appear to be swapped at places. A careful copy editing is needed.

Reviewer #2:

The revised paper addresses many of the shortcomings and is now technically of an acceptable standard. However, I am underwhelmed by the significance or novelty of the work.

- As I mentioned in the original review, there little novelty is identifying immediate late genes whose induction is delayed because their mRNAs have a long half-life. The authors do mention Hao et al 2009 who showed this elegantly many years ago, also using ectopic gene constructs in which mRNA half-life-determining elements were switched. They were neither the first, nor the last.

- Since the original submission, there is now less novelty in fitting kinetic models to mRNA measurements. Regev showed that mRNA half-life is important for gene induction, Batchelor

showed this for p53 targets also, and a new paper by Hoffman also shows this for NF-kB and IRF target genes.

- As I mentioned in the original review, the biological significance of the work remains unclear as the authors do not consider whether the protein products of ILGs may mediate the physiologic distinction between proliferation and differentiation, or even the responses of EGF vs NGF. The fact that some IEG protein products are stable (e.g. fos), as shown in the revised manuscript, undercuts the biological significance.

1st Revision - authors' response

29 March 2017

Response to Editorial comments

Taken together, we think that additional experiments are not mandatory for the acceptance of the manuscript. Of course, we would not be opposed to the inclusion of such analyses if you have already performed them or feel inclined to do so.

As detailed below we feel that establishing combinatorial multi-knockout experiments are not feasible within the scope of our study.

In line with the comment of reviewer #1, we would ask you to include a balanced discussion of previous studies on this topic.

We agree that our previous manuscript (which had no separate discussion section) did not adequately discuss our findings in relation to previous findings, and we now include an extensive discussion highlighting the relevant literature.

Moreover, reviewer #1 raises concerns regarding minor mistakes in the text, which should be addressed in a revision.

We apologise for these mistakes and carefully edited the manuscript in that respect.

Reviewer #1 cross-commenting:

I agree with reviewer 2's point that it is well established in the literature that mRNA half-life governs the kinetics of expression. However, my perspective is that this manuscript's value lies in 1) demonstrating that this concept applies to the ERK pathway, which despite being one of the most well-studied gene regulatory pathways has not been studied in this regard and 2) providing a comprehensive set of data on mRNA kinetics that will be useful for others working on this pathway. Thus, I think that the concerns about novelty are not sufficient to preclude publication and can be addressed with a balanced discussion of the literature. With regard to biological significance, I think the authors have made at least a reasonable effort to investigate this in Fig. EV7. While I agree that it would be very interesting to determine which genes underlie the proliferation/differentiation switch in the classic PC12 system, it seems to me that the multiplex nature of the system would put the complexity of the experiments required out of the scope of the study. (i.e., would it be possible to knock down enough ILGs to shift the NGF response from differentiation to proliferation? Would knockdown experiments even be sufficient, or would the steady state levels of multiple genes need to be tuned experimentally?) Other papers in this vein (for example, the Porter and Batchelor paper) have also found it difficult to ascribe defined biological functions to classes of mRNAs with different kinetics. Nonetheless, if a defined set of experiments to convincingly address this point could be prescribed, I would support asking the authors to attempt it.

We thank Reviewer #1 for the supporting comments. In our new discussion section, we now clearly state that the relation between mRNA half-life and gene expression kinetics is well established in the literature and discuss our findings in light of the mentioned papers (cf. lines 385-432). Also, we agree that an experiment that demonstrates a direct link between the group of ILGs and cell fate decision would be out of scope of our study, especially since commitment will likely depend on multiple genes and therefore single-gene knockouts will not suffice to show this point, making it experimentally too difficult.

Reviewer #1:

The authors have addressed all of our concerns adequately, and the added data on mRNA half-life strengthens the manuscript overall. However, there remain a number of confusing errors, in particular figure labeling issues for Fig 5 and Fig EV5 which appear to be swapped at places. A careful copy editing is needed.

We thank Reviewer #1 for the supporting comments and corrected the labelling issues for Fig 5 and EV5. We also double-checked all figure references across the entire manuscript.

Reviewer #2:

The revised paper addresses many of the shortcomings and is now technically of an acceptable standard. However, I am underwhelmed by the significance or novelty of the work.

- As I mentioned in the original review, there little novelty is identifying immediate late genes whose induction is delayed because their mRNAs have a long halflife. The authors do mention Hao et al 2009 who showed this elegantly many years ago, also using ectopic gene constructs in which mRNA halflife-determining elements were switched. They were neither the first, not the last.

- Since the original submission, there is now less novelty in fitting kinetic models to mRNA measurements. Regev showed that mRNA halflife is important for gene induction, Batchelor showed this for p53 targets also, and a new paper by Hoffman also shows this for NF-kB and IRF target genes.

We thank Reviewer #2 for highlighting the additional references and we included them in our new discussion section (cf. lines 385-432), where we stress their contributions and discuss our findings in relation.

- As I mentioned in the original review, the biological significance of the work remains unclear as the authors do not consider whether the protein products of ILGs may mediate the physiologic distinction between proliferation and differentiation, or even the responses of EGF vs NGF. The fact that some IEG protein products are stable (e.g. fos), as shown in the revised manuscript, undercuts the biological significance.

In agreement with Reviewer #1, we think that it is out of scope to establish a multi-knockout cell line to directly link protein products of ILGs to cell fate decision.

2nd Editorial Decision

04 April 2017

Thank you again for sending us your revised manuscript. We are now satisfied with the modifications made and I am pleased to inform you that your paper has been accepted for publication.

MOLECULAR SYSTEMS BIOLOGY

Corresponding Author Name: Nils Blüthgen
Manuscript Number: MSB-17-7554